# Beta-Glucan modulates monocyte plasticity and differentiation capacity to mitigate DSS-induced colitis

Yinyin Lv[1,2,3†], Yanyun Fan[1,2,3†], Qingxiang Gao[4†], Qiongyun Chen[2,3,5†], Yiqun Hu[1,2,3], Lin Wang[1,2,3], Huaxiu Shi[1,2,3], Ermei Chen[1,2,3], Qinyu Xu[1,2,3], Ying Cai[1,2,3], Qingqi Fan[4], Linying Li[4], Dan Du[6], Jianlin Ren[1,2,3]*, Shih-Chin Cheng[3,4]*, Hongzhi Xu[1,2,3]*

[1]Department of Gastroenterology, The National Key Clinical Specialty, Zhongshan Hospital of Xiamen University, School of Medicine, Xiamen University, Xiamen, China; [2]Clinical Research Center for Gut Microbiota and Digestive Diseases of Fujian Province, Xiamen Key Laboratory of Intestinal Microbiome and Human Health, Xiamen, China; [3]Department of Digestive Disease, Institute for Microbial Ecology, School of Medicine, Xiamen University, Xiamen, China; [4]State Key Laboratory of Cellular Stress Biology, School of Life Sciences, Faculty of Medicine and Life Sciences, Xiamen University, Xiamen, China; [5]Department of Gastroenterology, Taikang Xianlin Drum Tower Hospital, Affiliated Hospital of Medical School, Nanjing University, Nanjing, China; [6]State Key Laboratory of Cellular Stress Biology, Department of Gastroenterology, Zhongshan Hospital of Xiamen University, School of Medicine, Faculty of Medicine and Life Sciences, Xiamen University, Xiamen, China

**\*For correspondence:**
jianlin.ren@126.com (JR);
jamescheng@xmu.edu.cn (S-CC);
xuhongzhi@xmu.edu.cn (HX)

[†]These authors contributed equally to this work

## eLife Assessment

This study presents **compelling** evidence supporting the therapeutic potential of trained immunity in Colitis. The study is **important** for the field of trained immunity and is a welcome addition to the focus issue on trained immunity.

**Abstract** Trained immunity involves the reprogramming of innate immune cells after an initial exposure, resulting in heightened inflammatory responses to subsequent stimuli and enhanced bactericidal capacity during infection. However, this pro-inflammatory state could also exacerbate chronic conditions like inflammatory bowel disease (IBD), which is characterized by persistent inflammation and microbial imbalance. It remains unclear how trained immunity influences IBD pathogenesis and whether it can be harnessed therapeutically. In our study, pretreatment with β-glucan reprogrammed bone marrow hematopoietic progenitors and peripheral monocytes, inducing a profound shift in monocyte plasticity and significantly reducing the severity of dextran sulfate sodium (DSS)-induced colitis. Adoptive transfer of bone marrow or peripheral monocytes from β-glucan-trained mice into naive mice conferred robust protection against colitis, demonstrating that this protective effect is transferable. Trained mice also displayed improved clearance of intestinal bacterial infections. Single-cell RNA sequencing revealed an expansion of reparative Cx3cr1[+] macrophages derived from Ly6C[hi] monocytes, correlating with accelerated colonic epithelial regeneration. Collectively, these findings reveal how β-glucan-induced trained immunity modulates monocyte differentiation to ameliorate experimental colitis, highlighting the potential of harnessing trained immunity as a therapeutic strategy to recalibrate innate immune responses and restore gut homeostasis in IBD, shedding light for future clinical applications.

## Introduction

Inflammatory bowel disease (IBD), encompassing ulcerative colitis (UC) and Crohn's disease (CD), is a chronic gastrointestinal inflammatory disorder associated with significant morbidity (*Kobayashi et al., 2020*; *Dolinger et al., 2024*). The intestinal mucosa is continuously exposed to a diverse microbial ecosystem, including fungi, bacteria, and viruses (*Zhang et al., 2022a*). In IBD, the delicate balance between the immune system and the microbiota is disrupted, leading to uncontrolled inflammation (*Abraham and Medzhitov, 2011*; *Jostins et al., 2012*). This dysregulation is further exacerbated by genetic factors, such as loss-of-function mutations in NOD2, which impair antimicrobial peptide production, increase susceptibility to intestinal infections, and decrease IL-10 production by monocytes (*Wehkamp et al., 2004*; *Watanabe et al., 2004*; *Noguchi et al., 2009*).

Trained immunity, a phenomenon where innate immune cells exhibit enhanced responses to secondary challenges, has emerged as a promising therapeutic avenue for various infectious diseases and cancer (*Netea et al., 2016*; *Netea et al., 2020*; *Ding et al., 2023*; *Jeyanathan et al., 2023*; *Zhu et al., 2024*). However, its role in chronic inflammatory conditions like IBD remains largely unexplored. While trained immunity can enhance pathogen clearance, it also involves the upregulation of pro-inflammatory cytokines, such as TNF, IL-1β, and IL-6(14), therefore, the role of trained immunity in IBD remained elusive.

This study aimed to investigate the therapeutic potential of trained immunity in a mouse model of colitis. We hypothesized that inducing trained immunity could enhance microbial control, thereby preserving intestinal integrity and ultimately mitigating disease severity. To test this, we employed β-glucan (BG), a prototypic trained immunity inducer, and assessed its impact on dextran sulfate sodium (DSS)-induced colitis. Our findings demonstrate that BG pretreatment significantly alleviates colitis in mice. Mechanistically, BG induced central immunity in the hematopoietic compartment, leading to the generation of 'trained' monocytes in the periphery. Both bone marrow transplantation from trained donors and adoptive transfer of trained monocytes conferred protection against DSS-induced colitis. Furthermore, single-cell RNA-seq revealed that BG-induced trained immunity promotes the expansion of reparative Cx3cr1 intestinal macrophages derived from Ly6C$^{hi}$ monocytes, thereby facilitating epithelial regeneration. In summary, our data suggest that harnessing trained immunity represents a promising therapeutic approach for IBD treatment.

## Results

### β-glucan (BG) pretreatment ameliorates DSS-induced colitis

To investigate the impact of trained immunity on colitis, mice were pretreated with BG, a potent inducer of trained immunity. Successful induction of BG-induced trained immunity (BGTI) was confirmed by the enhanced resistance to *Staphylococcus aureus* infection in BG-treated mice (*Figure 1—figure supplement 1A and B*), consistent with previous findings (*Cheng et al., 2014*). Importantly, BG-pretreatment did not induce spontaneous intestinal inflammation (*Figure 1—figure supplement 1C and D*). Following BGTI induction, mice were subjected to DSS-induced colitis. (*Figure 1A*). BG pretreatment significantly ameliorated disease severity, as indicated by the reduced body weight loss (*Figure 1B* and *Figure 1—figure supplement 1E*), colon shortening (*Figure 1C* and *Figure 1—figure supplement 1F and G*), and diminished mucosal inflammation observed via endoscopic imaging (*Figure 1D*). Histological analysis demonstrated lower histopathologic scores (*Figure 1E and F*). Additionally, BG-pretreated mice exhibited enhanced intestinal epithelial barrier integrity, as evidenced by the increased expression of tight junction proteins Zo-1 and Occludin (*Figure 1G and H*), and decreased circulating FITC-dextran levels (*Figure 1I*).

Acknowledging the long-lasting effects of BGTI (*Kalafati et al., 2020*), we further examined its long-term protective effects against DSS-induced colitis. Despite peripheral myeloid cells and monocytes returning to homeostasis 4 weeks post-BG treatment (*Figure 1—figure supplement 2A and B*), mice remained protected against DSS-induced colitis as demonstrated by body weight change and colon shortening (*Figure 1J and K*), and improved histological outcomes (*Figure 1—figure supplement 2C and D*). This BG-induced protection persisted for up to 7 weeks (*Figure 1—figure supplement 2E and F*). These findings indicate that BG pretreatment effectively ameliorates DSS-induced colitis in mice, emphasizing the therapeutic potential of trained immunity in IBD management.

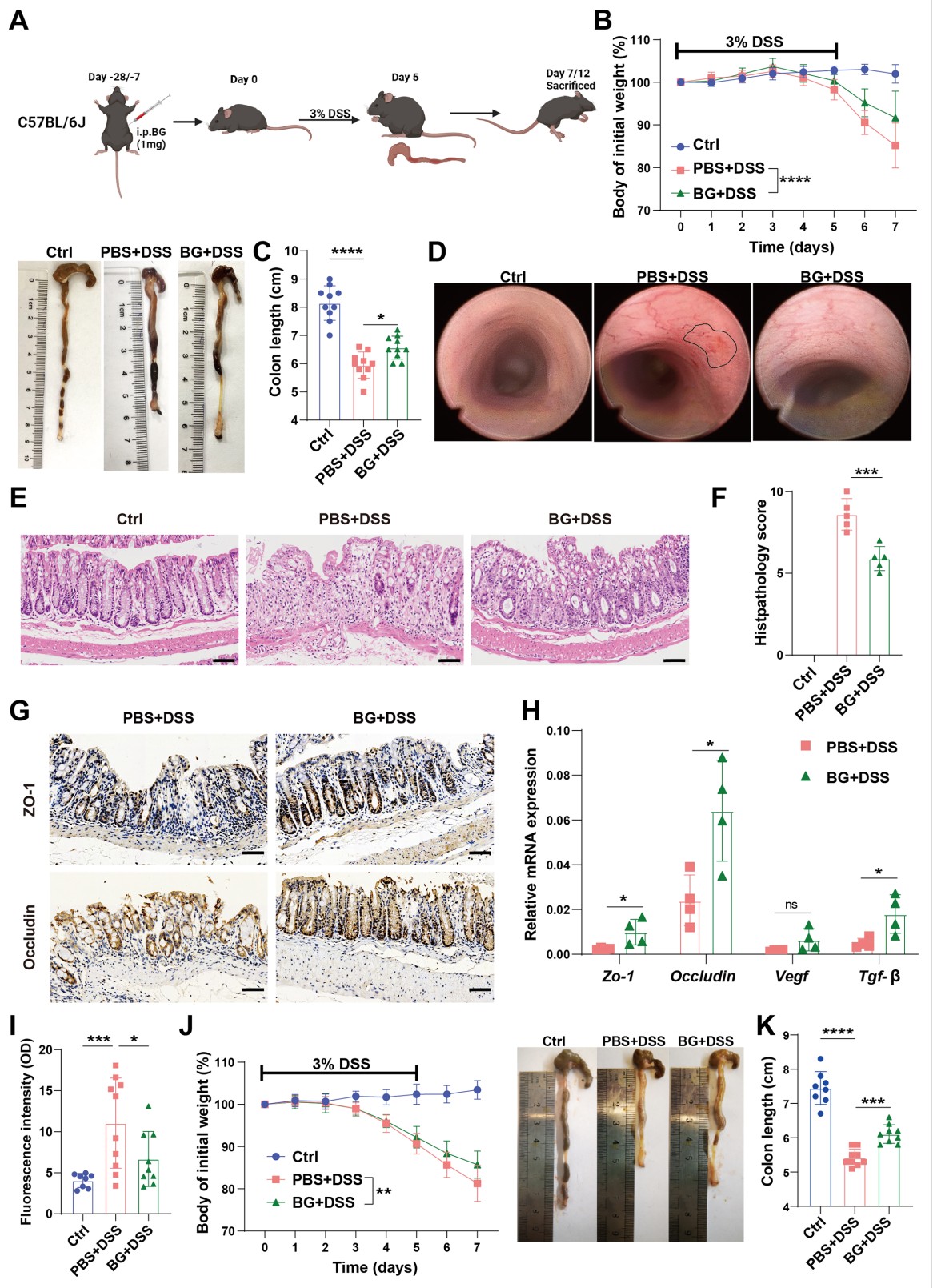

**Figure 1.** β-glucan (BG) pretreatment ameliorates dextran sulfate sodium (DSS)-induced colitis. (**A**) Schematic representation of BG-induced trained immunity and DSS colitis model. (**B**) Body weight change curve of mice pretreated with BG for 1 week, followed by colitis induction with 3% DSS (n=14–15). (**C**) Colon length changes in colitis mice (n=10). (**D**) Endoscopic images displaying mucosal damage. (**E, F**) H&E staining and histological scoring. Scale bars: 100 μm. (**G, H**) Expression levels of tight junction and repair. (**I**) FITC-dextran assay assessing intestinal barrier function. (**J**) Body weight

*Figure 1 continued on next page*

**Figure 1 continued**

change curve of mice pretreated with BG for 4 weeks, followed by colitis induction with 3% DSS (n=8). (**K**) Colon length changes in colitis mice (n=8–9). Data are presented as mean ± SD. Statistical significance: *p<0.05, **p<0.01, ***p<0.001, ****p<0.0001. ns, not significant.

The online version of this article includes the following figure supplement(s) for figure 1:

**Figure supplement 1.** β-glucan (BG) pretreatment ameliorates dextran sulfate sodium (DSS)-induced colitis.

**Figure supplement 2.** β-glucan (BG) pretreatment 4 or 7 weeks ameliorates dextran sulfate sodium (DSS)-induced colitis.

## BG ameliorates colitis via enhanced myeloid cell activation independent of adaptive immunity

Given that BGTI has been reported to function independently of adaptive immunity (*Netea et al., 2011*; *Quintin et al., 2012*), we evaluated the effects of BG pretreatment in *Rag1* knockout mice (*Figure 2—figure supplement 1A–1D*). Notably, BG pre-treatment remained effective in *Rag1⁻/⁻* mice, as evidenced by reduced body weight loss, colon shortening, and improved histological outcomes (*Figure 2—figure supplement 1E–1H*). These findings confirm that BGTI-mediated protection is independent of adaptive immune responses, consistent with previous findings demonstrating that adaptive immunity is dispensable for trained immunity.

To gain a comprehensive understanding of the molecular mechanisms underlying BG-mediated protection, we performed RNA sequencing on colonic samples collected at multiple time points following DSS administration. WGCNA identified the MEturquoise module emerged as the core functional module, containing 5015 genes whose expression exhibited significant co-regulation following BG intervention (*Figure 2—figure supplement 2A*). And differential gene expression of genes within this module was most significant on day 7 post-DSS treatment when comparing the BG-treated mice to PBS controls (*Figure 2—figure supplement 2B*). GO analysis revealed significant enrichment in pathways related to chemotaxis and leukocyte migration (*Figure 2A*). KEGG pathway enrichment analysis demonstrated significant activation of cellular processes involved in phagocytic function (*Figure 2B*) Additionally, the enrichment observed in WGCNA was further corroborated by whole-genome analysis of day 7 colitis samples, which also identified significant enrichment of pathways associated with chemotaxis, leukocyte migration, and phagocytic function (*Figure 2—figure supplement 2C and D*). Multiple analytical approaches concurrently indicated that BG pretreatment not only enhances the recruitment of immune cells but also promotes their functional activation, particularly with respect to phagocytic capabilities.

To further investigate how BG pretreatment reprograms of intestinal myeloid cells, we performed single-cell RNA-seq to characterize intestinal leukocytes populations on day 7 post-DSS treatment. Unbiased clustering analysis identified multiple clusters of intestinal immune cell subsets, including monocytes/macrophages (*Ly6c2*, *Ccr2*, and *Adgre1*), dendritic (DC) (*Cst3* and *H2-Aa*), neutrophils (*S100a8/a9* and *Ly6g*), B (*Cd79a Cd79b* and *Cd19*), Pre B (*Myl4* and *Mme*), CD4 T (*Cd3d* and *Cd4*), CD8 T (*Cd3d* and *Cd8a*), natural killer (*Nkg7*), ILC2 (*Gata3* and *Il4*), and ILC3 (*Rorc* and *Il22*) cells (*Figure 2C* and *Figure 2—figure supplement 2E*). Importantly, the ratio of monocytes/macrophages and neutrophils was elevated in the BG group (*Figure 2D*).

Single-cell RNA-sequencing data further demonstrated enhanced expression of innate pathogen recognition receptors (*Syk*, *Nod1*, *Nod2*, *Tlr2*, and *Tlr4*), efferocytosis (*Cd300lf*, *Axl*, *Mertk*, *Itgal*, and *Itgb7*), cytokines (*Il1b*, *Il1rn*, *Vegfa*, and *Tgfb1*), chemokines (*Ccl2*, *Ccl6*, *Ccl9*, *Cxcl9*), and chemokine receptors (*Ccr1*, *Ccr2*, *Cxcr4*) in intestinal monocytes/macrophages (*Figure 2E*). AUC and KEGG analysis demonstrated significant enrichment pathways related to activation of innate immune response and phagocytosis in monocytes/macrophages from BG-treated mice (*Figure 2F and G*). These findings, corroborated by WGCNA analysis, strongly suggest that BG pretreatment enhanced the recruitment and activation of myeloid cells, particularly monocytes/macrophages, in the inflamed colon, thereby improving disease outcomes.

## BG-trained bone marrow monocytes protected against colitis via the Ccl2-Ccr2 axis

Drawing upon single-cell transcriptome analysis that reveals the proportional expansion and functional reprogramming features of colonic monocyte/macrophage populations, in conjunction with the observation of significantly upregulated peripheral monocytes following BG pretreatment as depicted

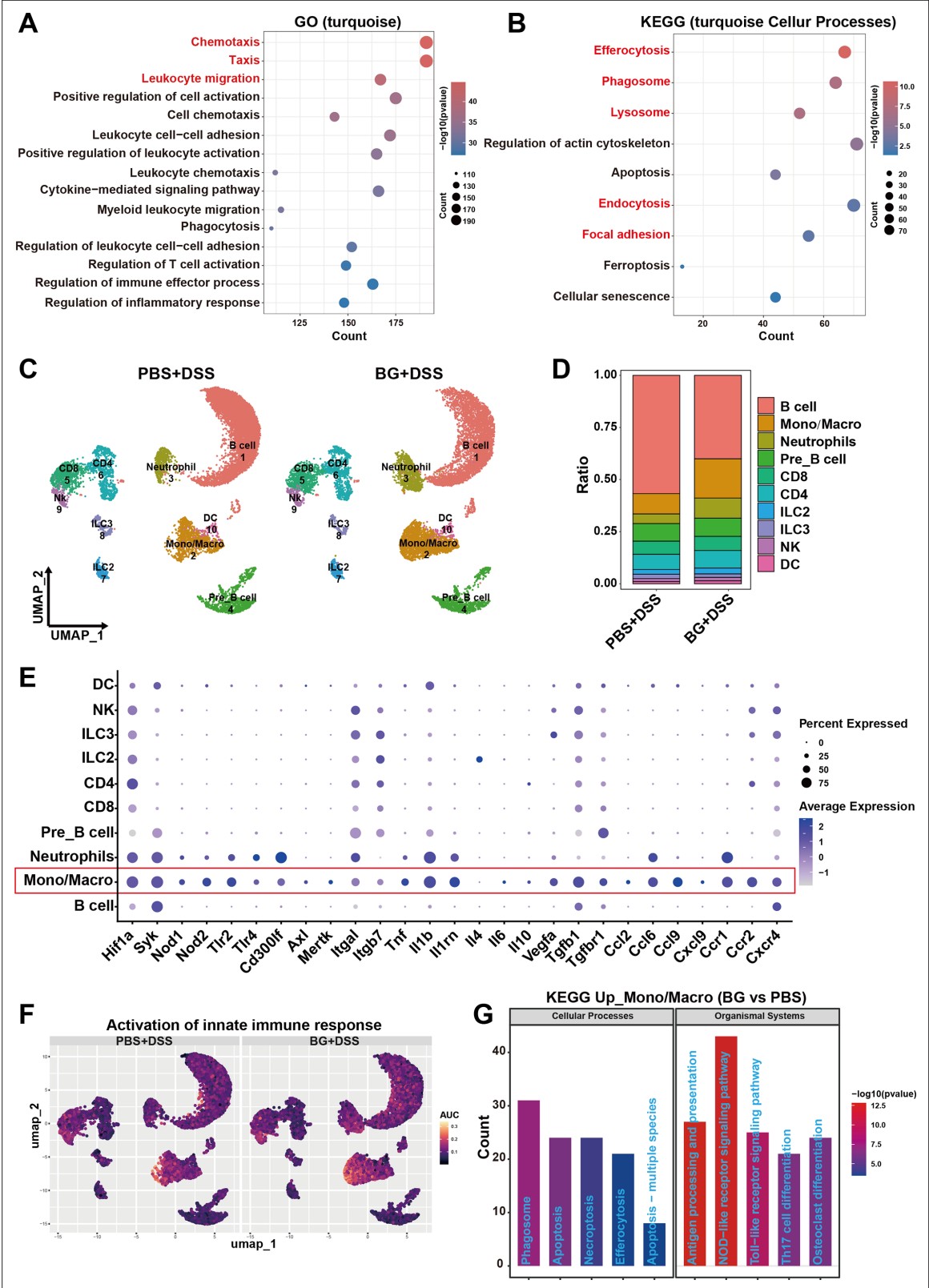

**Figure 2.** β-glucan (BG) ameliorates colitis by enhancing myeloid cell activation. RNA sequencing of colon tissue at different time points of colitis. (**A**) GOBP and (**B**) KEGG pathway analyses of genes in the MEturquoise module. Single-cell RNA sequencing analysis of CD45+ cells in the colon on day 7 of colitis after 1 week of BG pretreatment. (**C**) UMAP plot of LP CD45+ cells. (**D**) Cell ratio distribution from scRNA-seq data. (**E**) Dot plots showing representative DEGs between LP CD45+ cells. (**F**) AUC scores for selected pathways. (**G**) KEGG pathway analysis of genes upregulated in the monocyte-

*Figure 2 continued on next page*

*Figure 2 continued*

macrophage lineage. DEGs, differentially expressed genes. LP, lamina propria. UMAP, uniform manifold approximation, and projection. AUC, area under the curve.

The online version of this article includes the following figure supplement(s) for figure 2:

**Figure supplement 1.** β-glucan (BG) pretreatment ameliorates colitis is independent of adaptive immunity.

**Figure supplement 2.** β-glucan (BG) ameliorates colitis by enhancing myeloid cell activation.

in *Figure 1—figure supplement 2B*. A hypothesis was formulated that BG exerts protective effects against colitis via monocytes. To experimentally validate this hypothesis, we evaluated the role of monocytes in *Ccr2* KO mice. Circulating Ly6C$^{hi}$ monocytes were diminished in *Ccr2$^{-/-}$* mice (*Figure 3—figure supplement 1A and B*), corroborating the pivotal role of the Ccl2-Ccr2 axis in monocyte egression from the bone marrow into circulation (*Tsou et al., 2007*; *Geng et al., 2022*). Notably, BG pretreatment failed to protect *Ccr2$^{-/-}$* mice from DSS-induced colitis, as evidenced by the lack of significant differences in body weight loss, colon length, and epithelial permeability (*Figure 3A–E*).

In view of the fact that BG induces myelopoiesis through the modulation of hematopoietic stem and progenitors cell compartments in the bone marrow via central trained immunity (*Mitroulis et al., 2018*; *Chavakis et al., 2019*). We assessed whether transplantation of bone marrow cells from BG-pre-treated donors could confer resistance to DSS-induced colitis in naïve recipients (*Figure 3F*). At 6 weeks post-transplantation, circulating myeloid cells were predominately derived from donor mice as indicated by the CD45.1 marker on circulating mononuclear cells (*Figure 3—figure supplement 1C*). Following DSS treatment, mice receiving bone marrow cells from BG-pretreated donors exhibited significantly reduced body weight loss (*Figure 3G*). However, no significant difference in colon length was observed (*Figure 3H* and *Figure 3—figure supplement 1D*). Meanwhile, the percentage of circulating CD11b$^+$ myeloid cells, neutrophils, and Ly6C$^{hi}$ monocytes were increased (*Figure 3I and J*, *Figure 3—figure supplement 1E*).

To further assess the role of monocyte, we adoptively transferred bone marrow Ly6C$^{hi}$ monocytes sorted from control or BG-trained donor mice into *Ccr2$^{-/-}$* recipient mice, which were then subjected to DSS treatment (*Figure 3—figure supplement 1F and G*). Mice received BG-trained monocytes exhibited reduced weight loss and colon shortening (*Figure 3K and L*). These findings suggest that BG-trained Ly6C$^{hi}$ Ccr2$^+$ monocytes play a pivotal role in the alleviation of colitis.

## BG-trained monocytes enhance innate immune activation and microbial control

To explore the heterogeneity and functional roles of monocytes and macrophages, we delved deeper into the scRNA-seq data. By examining the expression of key signature genes, including *Ly6c2*, *Ccr2*, *H2-Ab1*, *Runx3*, *Itgax*, *Adgre1*, *Cx3cr1*, and *Cd209a*, we identified eight distinct immune cell subsets: three monocyte subsets (Mono1-3) characterized by high *Ly6c2* and *Ccr2* expression, four macro-phage subsets (Macro1-4) defined by expression of *H2-Ab1*, *Runx3*, *Itgax*, *Adgre1*, and *Cx3cr1*, and a dendritic cell subset (Cd209a$^+$DC) expressing integrin *Itgax* and C-type lectin *CD209a* (*Figure 4A* and *Figure 4—figure supplement 1A*). Among macrophage subsets, Macro1 highly expressed *Vegfa*, suggesting that may be a mucosal repair-promoting macrophage population. Macro2 was enriched with tissue-resident macrophage signature genes (*Runx3*, *Dtx4*, *Cx3cr1*, *Hes1*). In contrast, Macro3 and Macro4 deviated from the main cluster in cell clustering analysis, showing significant heterogeneity. Macro3 was characterized by high *Ighm* expression, while Macro4 showed elevated *Cd4* expression. Both Macro2 and Macro4 exhibited higher *Il10* expression than other monocyte/macrophage subsets, indicating their inflammation-regulatory functions.

The BG-treated group exhibited a significant increase in the proportions of Mono1, Mono2, Macro1, and CD209a$^+$DCs, whereas higher proportions of Mono3 and Macro3 were observed in the PBS group (*Figure 4B*). Given the strong enrichment of Mono1 and the reduction of Mono3 following BG treatment, we sought to further define the transcriptomic characteristics that differentiate Mono1 from Mono3. GO enrichment analysis revealed that Mono1 was significantly enriched in pathways regulating the activation of innate immune response (*Figure 4—figure supplement 1B*). KEGG pathways analysis further identified enrichment in NOD-like receptor (NLR) signaling, Toll-like receptor (TLR) signaling, and phagocytosis (*Figure 4C*). AUC analysis further confirmed the TLR and NLR signaling

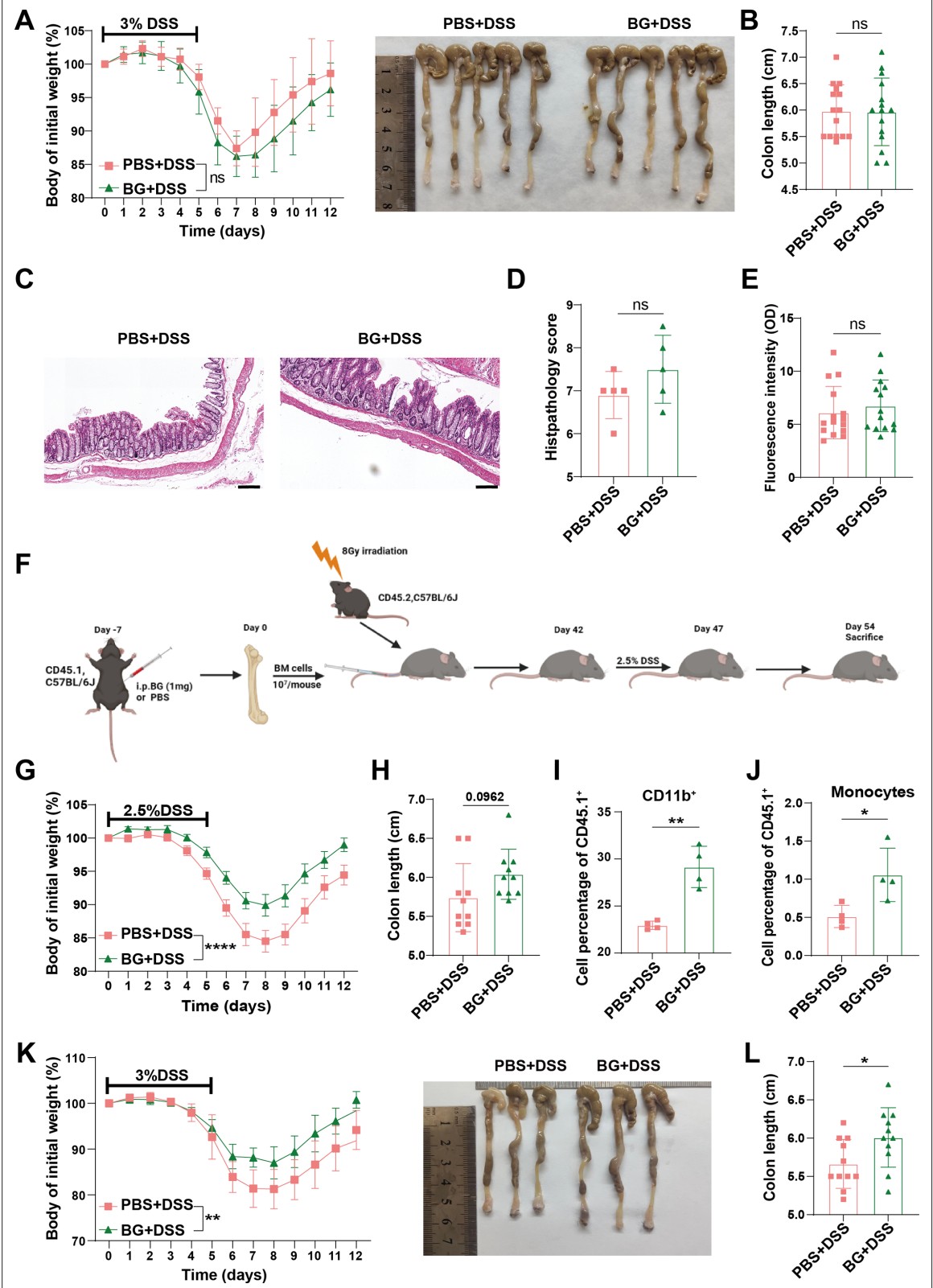

**Figure 3.** β-glucan (BG) trained bone marrow monocytes protected against colitis via the Ccl2-Ccr2 axis. *Ccr2*[-/-] mice pretreated with BG for 1 week followed by induction of colitis with 3% dextran sulfate sodium (DSS). (**A**) Changes in body weight (n=10). (**B**) Colon length changes in *Ccr2*[-/-] colitis mice (n=15). (**C, D**) H&E staining and histological scoring, H&E. Scale bars: 100 μm. (**E**) FITC-dextran assay assessing intestinal barrier function. (**F**) Schematic representation of bone marrow transplantation from BG-pretreated CD45.1 mice to CD45.2 mice and colitis model. (**G**) Body weight change curve of

*Figure 3 continued on next page*

*Figure 3 continued*

colitis mice (n=10). (**H**) Colon length changes in colitis mice from (**G**). (**I, J**) Percentage of CD11b⁺ (**I**) and Ly6Cʰⁱ monocytes (**J**) were analyzed by flow cytometry in peripheral blood. (**K, L**) Body weight change curve (**K**) and colon length changes of *Ccr2⁻ᐟ⁻* mice receiving Ly6Cʰⁱ monocyte adoptive transfer (**L**) (n=11). Data are presented as mean ± SD. Statistical significance: *p<0.05, **p<0.01, ****p<0.0001. ns, not significant.

The online version of this article includes the following figure supplement(s) for figure 3:

**Figure supplement 1.** β-glucan (BG) trained bone marrow monocytes protected against colitis via the Ccl2-Ccr2 axis.

**Figure supplement 2.** Adoptive transfer of β-glucan (BG)-trained monocytes ameliorates experimental colitis.

pathways were significantly upregulated in Mono1 and Mono2 subsets of the BG-pretreated group (*Figure 4—figure supplement 1C and D*). Additionally, antimicrobial humoral response pathway was also upregulated in these two subclusters (*Figure 4—figure supplement 1E*). Cooperative activation of the pattern recognition receptor (PRR) signaling axis and antimicrobial humoral response significantly promotes the synthesis of antimicrobial peptides (*Muniz et al., 2012*; *Duarte-Mata and Salinas-Carmona, 2023*). BG pretreatment significantly upregulated genes encoding pattern recognition receptors (*Nod1, Nod2, Tlr2, Tlr4*) and antimicrobial effector molecules (*S100a8, S100a9, Lcn2, and Defb1*) in the colon on day 7 of colitis (*Figure 4D*). Furthermore, antimicrobial genes *S100a8* and *Defa1* were highly expressed in the BG group as validated by qPCR analysis (*Figure 4—figure supplement 1F*). These genes upregulation suggest a potential link between trained immunity and enhanced microbial control. Consistently, gene signatures associated with bacterial defense responses were prominently enriched in the BG-treated group (*Figure 4E*).

To investigate the epigenetic basis for this sustained transcriptional reprogramming, we reanalyzed publicly available ATAC-seq data from hematopoietic stem cells (HSCs) of BG-trained mice (CRA014389). This analysis revealed that BG training reshaped the epigenomic landscape, with pronounced chromatin opening at promoters and distal intergenic regulatory loci (*Figure 4—figure supplement 2A, B*). Functional annotation of these primed regions showed significant enrichment for immune processes, including leukocyte migration and chemotaxis (*Figure 4—figure supplement 2C*), as well as pathways like chemokine signaling (*Figure 4—figure supplement 2D*). Crucially, promoter-centric analysis highlighted the enrichment of defense response to bacterium (*Figure 4—figure supplement 2E*), and locus-specific examination confirmed BG-induced chromatin remodeling upstream of key antimicrobial effectors, including Gbp2, Gbp5, S100a8, and Nos2 (*Figure 4—figure supplement 2F*). This demonstrates that BG establishes a permissive epigenetic architecture in progenitor cells, priming the expression of immune and antibacterial programs.

Consistent with this primed epigenetic state, we observed a significant enrichment of the IFN-γ response pathway (*Figure 4—figure supplement 1G*) and a specific upregulation of Guanylate-binding protein (Gbp) genes (*Gbp2, Gbp3, Gbp5, Gbp7*) in the Mono1 and Mono2 subsets (*Figure 4F*). Previous studies have demonstrated that GBPs as intracellular effectors induced by IFNγ and LPS to promote antibacterial defense (*Sweet et al., 2025*; *Pilla et al., 2014*). GBP-deficient mice exhibit significantly increased susceptibility to Salmonella typhimurium infection (*Man et al., 2016*; *Santos and Broz, 2018*; *Meunier et al., 2014*). We then performed intestinal Salmonella typhimurium infection to assess whether BG pretreatment enhances microbial control in the gut (*Figure 4G*). As expected, BG pretreatment significantly protected mice from lethal Salmonella Typhimurium infection, suggesting that BG-induced trained immunity enhances mucosal defenses against gut microbial infections (*Figure 4H*).

Taken together, these results reinforced our bulk RNA-seq findings, indicating that BG pretreatment induces a specific monocyte subcluster with enhanced innate activation and phagocytic capacity, facilitating more efficient control of the breaching microbiome associated with DSS-induced leaky gut.

## BG-mediated reprogramming of myeloid differentiation trajectories balances inflammation and enhances mucosal repair in colitis

While BG-induced trained immunity is characterized by enhanced pro-inflammatory cytokine production and increased bactericidal capacity (*Moorlag et al., 2020*), excessive pro-inflammatory cytokines release may exacerbate colitis progression. To balance heightened cytokine production without inducing overwhelming immunopathology, we hypothesized that BG pretreatment might involve feedback loops that modulate the inflammation while preserving enhanced phagocytic capacity.

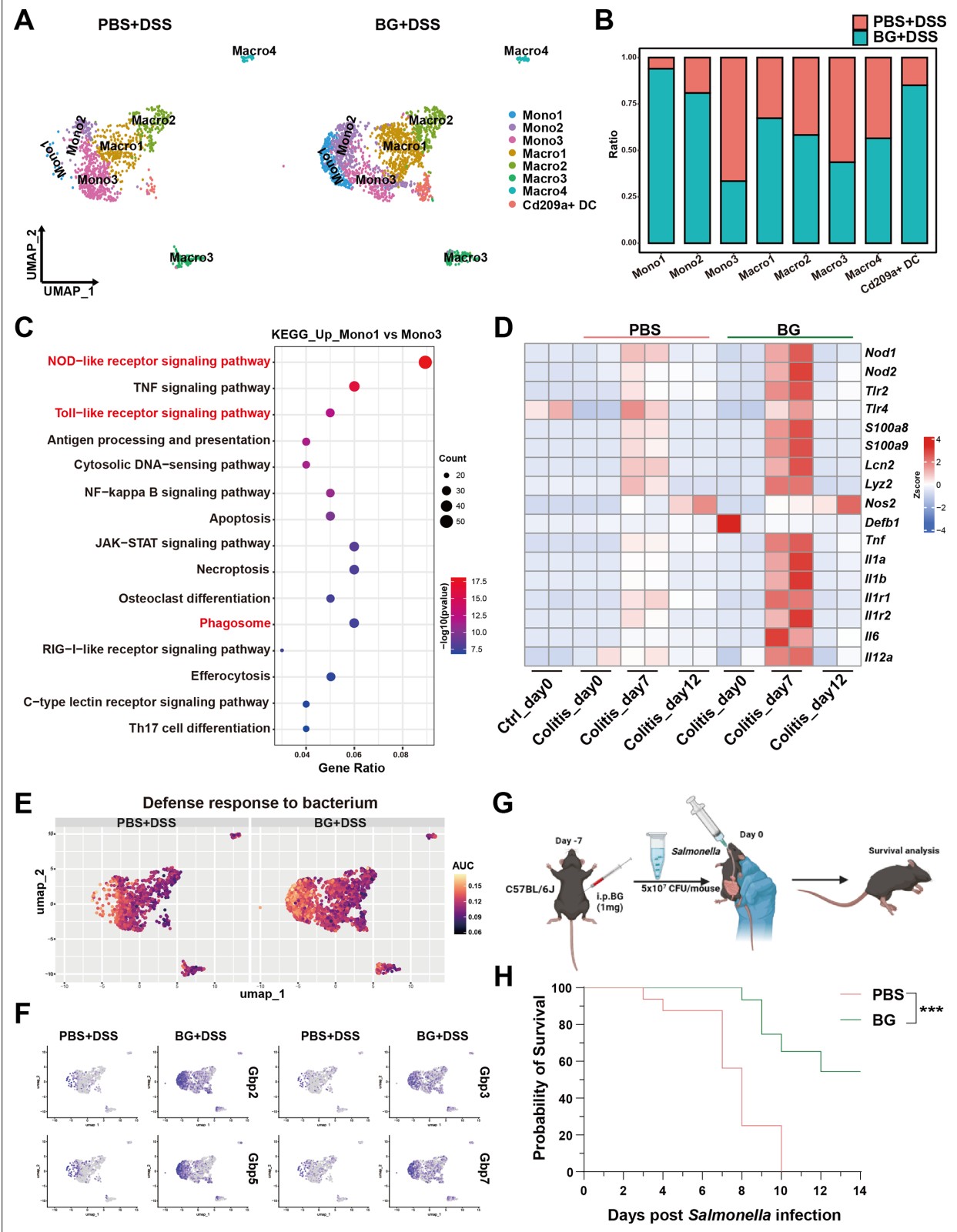

**Figure 4.** β-glucan (BG)-trained monocytes enhance innate immune activation and microbial control. (**A**) Uniform manifold approximation and projection (UMAP) and graphical visualization of the monocyte/macrophage lineage. (**B**) The ratio of monocyte/macrophage subsets. (**C**) KEGG pathway enrichment analysis of genes upregulated in monocyte 1 and monocyte 3. (**D**) Gene expression analysis at different time points of colitis, based on colon RNA sequencing. (**E**) Area under the curve (AUC) scores of selected pathways. (**F**) UMAP plots showing differential gene expression patterns.

*Figure 4 continued on next page*

*Figure 4 continued*

(**G**) Schematic representation *Salmonella* resistance after 1 week of BG pretreatment. (**H**) Survival curve of mice infected with *Salmonella* after 1 week of BG training. Statistical significance: ***$p<0.001$.

The online version of this article includes the following figure supplement(s) for figure 4:

**Figure supplement 1.** β-glucan (BG) pretreatment enhances innate immunity and phagocytic capacity.

**Figure supplement 2.** BG induced immune-related epigenomic remodeling in monocytes.

To investigate this, we analyzed previously obtained bulk colonic RNA-seq data, focusing on genes associated with anti-inflammatory and regulatory functions. On day 7 of colitis, BG pretreatment significantly upregulated genes encoding immunomodulatory factors (*Il4*, *Il10*, *Il11*, and *Il33*), signaling inhibitors (*Socs1*, *Socs3*), and differentiation regulators (*Csf1*, *Csf1r*, *Stat3*, and *Stat6*) (*Figure 5—figure supplement 1A*) The upregulation of *Socs1* and *Socs3* was further corroborated by scRNA-seq data, which showed increased expression of these genes in the Mono1 and Mono2 subclusters (*Figure 5—figure supplement 1B*). As members of the suppressor of cytokine signaling (SOCS) family, SOCS1 and SOCS3 inhibit excessive JAK-STAT signaling activation via negative feedback regulation (*Yuan et al., 2023*; *Zhao et al., 2023*; *Bidgood et al., 2024*). Moreover, coordinated activation of the monocyte/macrophage differentiation regulatory network (*Csf1r*, *Nr4a2*, *Irf8*, *Klf4*) suggests that BG may induce the differentiation of regulatory-phenotype macrophage subsets by reprogramming myeloid developmental trajectories, thereby modulating inflammatory responses.

To further investigate whether BG pretreatment reprograms monocyte-to-macrophage differentiation, we analyzed the day 7 RNA-seq data, which revealed significant enrichment of myeloid leukocyte differentiation pathway (*Figure 5—figure supplement 1C*). Additionally, Single-cell trajectory analysis further revealed that Macro1 and Macro2 differentiated from Mono2 and Mono3, respectively (*Figure 5A*). BG training significantly accelerated the differentiation kinetics of Mono3 into Macro1, which may underlie the reduced frequency of the Mono3 subset in the BG group. Coupled with the gene expression signatures in *Figure 4—figure supplement 1A*, Macro1 shares similarities with Macro2 in marker gene expression, and the differentiated Macro1 subset could further differentiate into the tissue-resident macrophage Macro2 subset (*Figure 5B*).

Furthermore, we observed that the expression levels of *Ccr2* and *Ly6c2* decreased from Mono1 to Macro2, while *H2-Ab1* and *Cx3cr1* progressively increased, particularly in macrophages (*Figure 5C*). Since Cx3cr1 upregulation is a hallmark of circulating monocytes migrating to the intestinal lamina propria and undergoing macrophage differentiation (*Li et al., 2021*), we used *CX3CR1-GFP* reporter mice to monitor the monocyte-to-macrophage transition in the DSS colitis model. In circulating leukocyte subsets, the total percentage of CD11b$^+$ myeloid cells was higher in BG-pretreated mice compared to controls (*Figure 5—figure supplement 2A–C*). A similar trend was observed in neutrophils (*Figure 5—figure supplement 2D and E*). While the initial monocyte percentage was higher in the BG group, no significant difference was observed the two groups on day 7 post-DSS treatment (*Figure 5D*), suggesting that circulating monocytes in BG-pretreated mice had extravasated into the intestinal tissue. Meanwhile, the percentage of colonic CD11b$^+$ myeloid cells and neutrophils was increased in the BG-pretreated group (*Figure 5—figure supplement 3A–C*).

We further defined monocyte/macrophage populations from P1 to P6 based on the expression of CD11b, CX3CR1-GFP, Ly6C, and MHCII. P1 to P3 were gated from CD11b$^+$&CX3CR1-GFP$^-$&Ly6G$^-$ cells. While P1 (Ly6C$^+$&MHCII$^-$) and P3 (Ly6C$^-$&MHCII$^+$) were comparable between groups, P2 (Ly6C$^+$&MHCII$^+$) was increased in BG-pretreated mice (*Figure 5E*). P4 to P6 are gating from CD11b$^+$&CX3CR1-GFP$^+$ cells and further separate by the expression of Ly6C and MHCII. P4 (Ly6C$^+$&MHCII$^-$) as infiltrating monocytes gaining CX3CR1 expression. P5 (Ly6C$^+$&MHCII$^+$) is regarded as intermediate transitory immature macrophage and P6 (Ly6C$^-$&MHCII$^+$) is considered as mature macrophage as it completely loss expression of Ly6C and highly express MHCII. BG pretreatment led to an increase in P5 level (*Figure 5F*). The increase of P2 and P5 in BG group in line with the decrease of monocytes at day 7 in circulation post DSS treatment, suggesting that BG-trained monocytes have higher capacity to infiltrate into the damaged colon and undergo macrophage differentiation.

Previous WGCNA analysis suggested that BG pre-treatment upregulated the focal adhesion pathway, which may enhance tissue repair mechanisms (*Figure 2B*). Consistent with this, we observed increased expression of genes involved in wound healing and tissue repair, such as *Mmp*-related

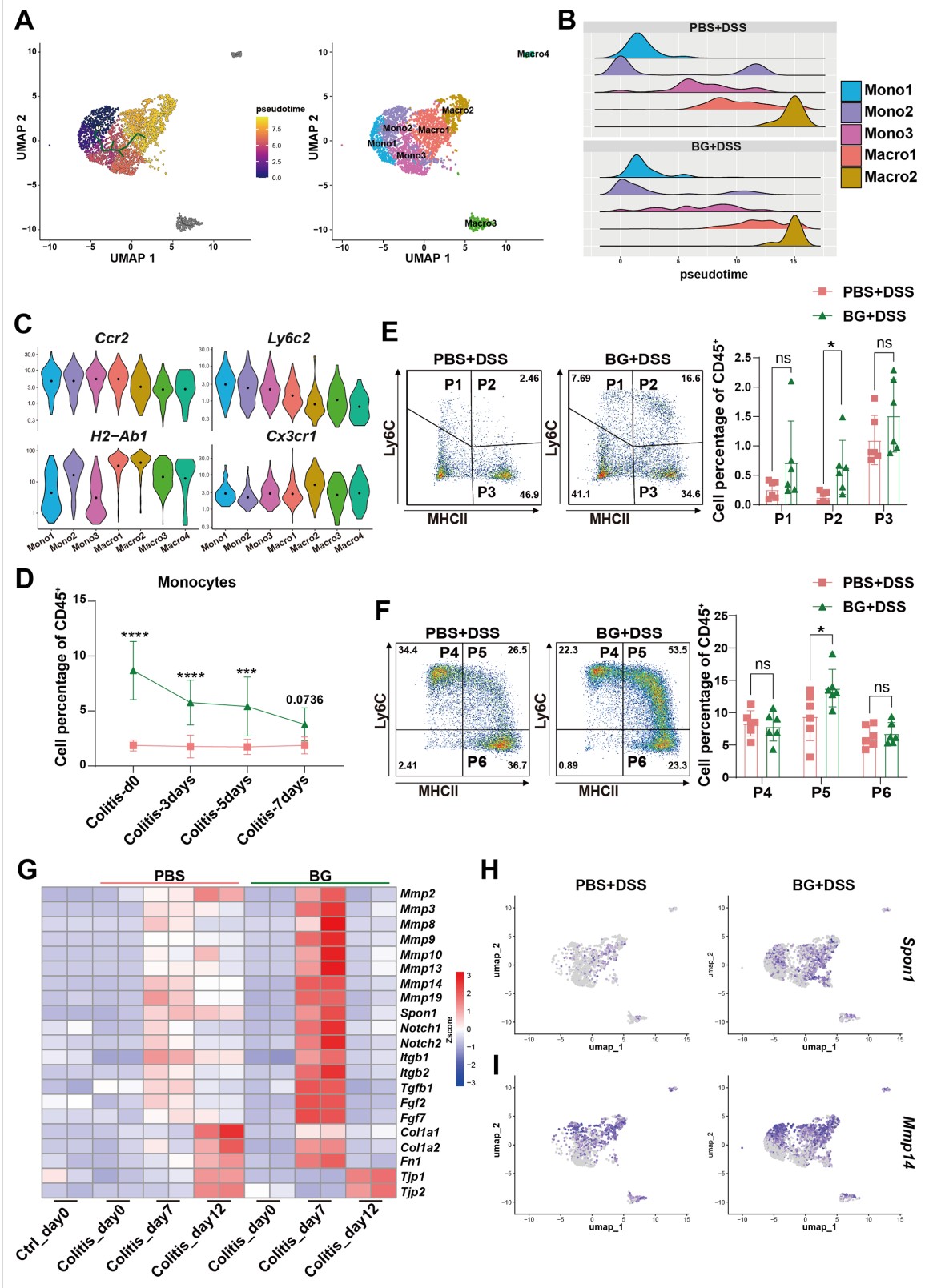

**Figure 5.** β-glucan (BG)-mediated reprogramming of myeloid differentiation trajectories balances inflammation and enhances mucosal repair in colitis. (**A**) Monocle 3 trajectory analysis of monocyte/macrophage subsets. (**B**) Ridgeline plot of monocyte/macrophage subsets. (**C**) Violin plots of surface marker gene expression in monocyte/macrophage subsets. (**D**) Percentage of monocytes in peripheral blood at different time points of colitis progression. (**E, F**) Colonic LPMCs were collected, and the percentages of monocyte/macrophage were analyzed on day 7 of colitis. (**G**) Expression of

*Figure 5 continued on next page*

Figure 5 continued

mucosal repair-related genes at different time points of colitis. (**H**) Gene expression analysis of the monocyte/macrophage lineage. Data are presented as mean ± SD. Statistical significance: *p<0.05, ***p<0.001; ****p<0.0001. ns, not significant.

The online version of this article includes the following figure supplement(s) for figure 5:

**Figure supplement 1.** β-glucan (BG)-induced trained immunity promotes monocyte differentiation.

**Figure supplement 2.** β-glucan (BG) training downregulates the proportion of peripheral monocytes during colitis development.

**Figure supplement 3.** β-glucan (BG) training upregulates the proportion of colonic monocytes/macrophages in colitis mice.

genes, *Spon1*, *Notch1*, *Notch2*, *Tgfb*, *Fgf1*, *Fgf2*, *Col1a1*, *Col1a2*, and *Fn1* in the BG-treated group on day 7, followed by a rapid decline by day 12. In contrast, the PBS group exhibited a much milder induction of these genes, with several genes remaining at comparable levels between day 7 and day 12 (*Figure 5G*). This led us to hypothesize that the upregulated monocytes and macrophages in the BG-treated group might exhibit enhanced expression of tissue repair-associated genes. As expected, the expression of *Spon1* and *Mmp14*, which promote the mucosal repair was increased in Mono2, Macro1, and Macro2 (*Figure 5H and I*). These findings suggest that BG pretreatment indeed induced monocyte and macrophage subsets with enhanced tissue repair capacity.

## Discussion

Trained immunity, a non-antigen-specific form of innate immunological memory, enables rapid host defense mobilization upon pathogen challenge (*Ziogas et al., 2023*; *Ochando et al., 2023*). In inflammatory bowel disease, gut microbial dysbiosis manifests as reduced α-diversity, diminished commensal taxa (e.g. *Faecalibacterium prausnitzii*), and pathobiont expansion (e.g. *Enterobacteriaceae*, *adherent-invasive Escherichia coli*), which may compromise intestinal barrier integrity and precipitate microbial translocation (*Sauceda et al., 2022*; *Schaubeck et al., 2016*; *Palmela et al., 2018*). Additionally, the long-term use of immunosuppressants and biologics in IBD patients elevates infection susceptibility, rendering the microbial-controlling properties of trained immunity a viable therapeutic strategy for IBD (*Kirchgesner et al., 2018*). However, Excessive or sustained immune activation could trigger chronic inflammatory cascades and provoke autoimmune tissue damage through aberrant immune recognition, as observed in conditions, such as rheumatoid arthritis and cardiovascular diseases (*Netea et al., 2016*; *Hu et al., 2022*; *Ospelt et al., 2011*; *Grigoriou et al., 2020*). However, the functional correlation between trained immunity and colitis remains unclear, and how to apply trained immunity to colitis treatment represents an urgently unresolved scientific question.

Here, we demonstrated that BG-induced trained immunity confers cross-pathogen protective effects using a lethal *Staphylococcus aureus* infection model, while failing to induce intestinal inflammation under conditions of an intact mucosal barrier. Further investigations revealed that BG effectively attenuated dextran sulfate sodium (DSS)-induced colitis upon secondary challenge. BG pretreatment conferred long-term protection against DSS-induced colitis, with benefits lasting for at least two months after the peripheral myeloid cell populations returned to baseline levels. Bone marrow transplantation experiments confirmed that BG-induced trained immunity persists within hematopoietic progenitors, leading to durable reprogramming of peripheral monocytes. This long-term effect is likely supported by epigenetic modifications in hematopoietic cells, as reported in previous studies (*Mitroulis et al., 2018*; *Tercan et al., 2021*). These findings highlight the potential of BG-induced trained immunity in establishing long-lasting immune memory, which could be leveraged to achieve sustained therapeutic effects in IBD.

The conventional understanding of IBD pathogenesis has long posited that abnormal immune activation is central to disease development and progression. This paradigm has predominantly focused on the role of adaptive immunity, particularly the dysregulation of Th17 and Treg functions over the past decades (*Liu et al., 2024*; *Kosinsky et al., 2024*; *Britton et al., 2019*). However, our results suggest that BG-induced protection is independent of adaptive immune responses. Even in *Rag1*[-/-] mice, which lack adaptive immunity, BG pretreatment significantly improved DSS-induced colitis outcomes. *Rag1*[-/-] mice retain ILCs (including ILC3s), which promote mucosal repair via IL-22 production (*Almeida and Belz, 2016*). While BG-induced ILC activation remains untested, our monocyte adoptive transfer experiments show monocytes alone alleviate colitis—suggesting a dominant, but

not exclusive, role for monocytes/macrophages. ILC contributions cannot be ruled out and warrant further investigation.

In DSS-induced colitis, epithelial barrier disruption facilitates microbiota translocation across the mucosal epithelium, triggering inflammatory responses mediated by innate cells (*Palmela et al., 2018*; *Hernández-Chirlaque et al., 2016*; *Panpetch et al., 2020*). We identified monocytes as key mediators of BG-induced protection against DSS-induced colitis, as demonstrated by experiments using *Ccr2*[-/-] mouse models and monocyte transfer studies. BG pretreatment enriched Ly6C[hi]CCR2[+] monocytes, which exhibited enhanced antimicrobial capabilities, as revealed by scRNA-seq. Ly6C[hi]CCR2[+] monocytes displayed higher levels of antimicrobial receptors like NOD2, TLR4, and GBP-related gene. Guanylate-binding protein (GBP) enhances *Salmonella* clearance by promoting inflammasome activation and intracellular bacterial elimination, particularly through disrupting *Salmonella*-containing vacuoles and inducting pyroptosis (*Man et al., 2016*; *Rupper and Cardelli, 2008*). Our finding revealed that BG pretreatment increased resistance to *Salmonella typhimurium* infection. As inflammation and infection is partially controlled, monocytes can differentiate into macrophage subsets with anti-inflammatory and tissue repair functions, secreting cytokines, such as IL-10, TGF-β, and VEGF to promote inflammation resolution and tissue repair (*Rigamonti et al., 2023*; *Wynn and Vannella, 2016*).

Previous studies have underscored the essential role of circulating Ly6C[hi] monocytes in replenishing intestinal macrophages (*De Schepper et al., 2019*; *Bain et al., 2013*). During colitis and infection, these monocytes are rapidly recruited to injured colonic tissues, where they differentiate into Cx3cr1[+] macrophages. We also observed that these monocytes exhibited enhanced migratory capabilities, enabling them to infiltrate inflamed tissues and differentiate into CX3CR1[+] macrophages. These macrophages contribute to mucosal repair by clearing pathogens, such as *Salmonella* and promoting epithelial regeneration (*Trebicka et al., 2015*). Single-cell trajectory analysis further showed that BG training promoted the differentiation of Macro1 into Macro2, which is consistent with the dynamic process of monocyte-to-mature macrophage conversion observed in CX3CR1-GFP reporter mice. Maintaining a balance between inflammatory monocytes, differentiated macrophages, and tissue-resident macrophages is critical for resolving inflammation, promoting tissue repair (*Li et al., 2021*; *Weber et al., 2011*). These findings suggest that BG not only augments monocyte antimicrobial functions but also facilitates their differentiation into macrophages, enhancing their ability to recognize, phagocytose, and eliminate bacteria while promoting tissue repair. This dual function likely strengthens the intestinal barrier, mitigates inflammation, and supports gut homeostasis.

Notably, our findings align with emerging evidence that trained immunity (TI) is not a uniform biological process but encompasses distinct functional programs shaped by the initiating stimulus and contextual cues—including immunoregulatory and tissue-reparative responses. Yang Li et al have identified discrete monocyte/macrophage subpopulations following TI induction, such as cytokine-chemokine dual-high (MCI) and chemokine-high (MC) subsets, which exhibit distinct pathway enrichments and disease associations (*Zhang et al., 2022b*). Consistent with this, β-glucan (BG), the TI inducer in our study, has been shown to drive a specific subset of macrophages with M2-like phenotypic traits (e.g. high CD206 expression, increased IL-10 secretion) while retaining potent antimicrobial capabilities (e.g. bacterial killing, pro-inflammatory cytokine release upon LPS stimulation)—reflecting an immunoregulatory TI program that balances pathogen clearance and inflammation resolution (*Leonhardt et al., 2018*). Furthermore, BG-induced TI has been reported to inhibit NLRP3 inflammasome activation via blocking K[+] efflux and mitochondrial ROS generation, highlighting its context-dependent anti-inflammatory potential in autoinflammatory diseases (*Camilli et al., 2020*)—a mechanism that may contribute to its protective effects in colitis by limiting excessive intestinal inflammation. Our findings of BG-induced protection against DSS colitis, characterized by enhanced microbial clearance and accelerated mucosal repair without exacerbating inflammation, represent a paradigmatic example of such context-dependent, reparative TI programs. This aligns with the broader understanding that TI can be tailored to elicit tissue-protective effects when triggered by appropriate stimuli, as demonstrated by BG's ability to prime myeloid cells for both antimicrobial defense and tissue repair (*Hajishengallis et al., 2019*). Together, these observations place our work within the growing framework of TI functional heterogeneity, emphasizing that the biological outcome of TI is dictated by the interplay between the inducing stimulus, target cell type, and local microenvironment—an insight that further clarifies the mechanism by which BG-induced TI confers colitis protection.

Immunology and Inflammation

In the clinical exploration of IBD treatment, novel therapies based on cellular regeneration (intestinal organoid transplantation) and fecal microbiota transplantation (FMT) are emerging as critical avenues to overcome the limitations of conventional pharmacotherapies (*Yui et al., 2012*; *Lopetuso et al., 2023*; *Haifer et al., 2023*). Given the efficacy of BG-induced trained immunity in alleviating DSS-induced colitis, this approach holds significant promise as a novel therapeutic strategy for IBD treatment. We transferred BG-preconditioned monocytes which significantly alleviated clinical symptoms, including weight loss and colon shortening, demonstrating the potential of cell-based therapies (*Figure 3—figure supplement 2A–C*).

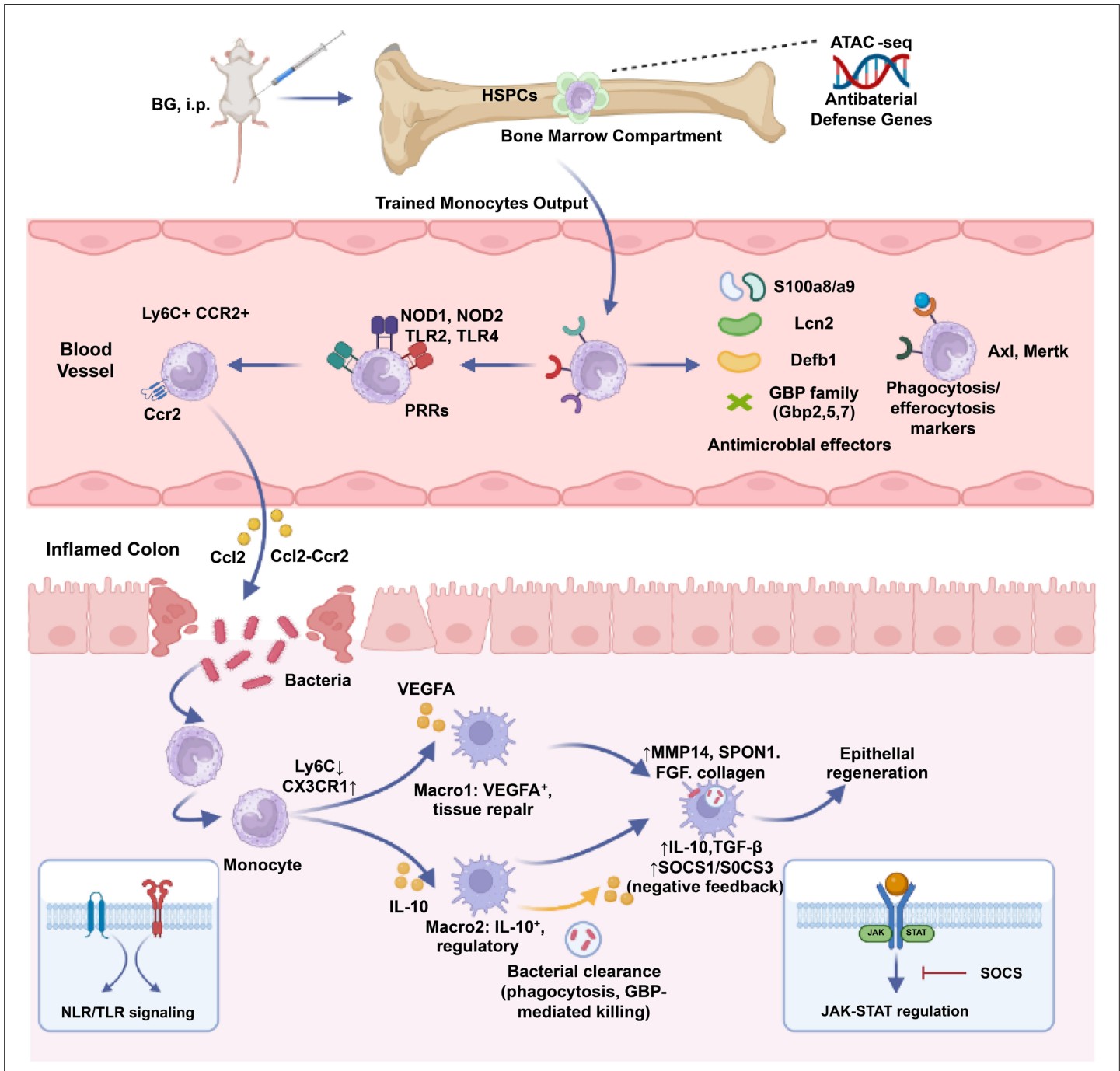

**Figure 6.** Graphical abstract.

In conclusion, BG confers long-term colitis amelioration through trained immunity by reprogramming bone marrow-derived monocytes. Mechanistically, BG enhances monocyte chemotaxis and antimicrobial functionality activation, enabling efficient microbial containment. Crucially, this trained state establishes self-regulating circuits, engaging in negative feedback regulation of excessive immune responses, while accelerated CX3CR1+ macrophage differentiation facilitates mucosal repair (*Figure 6*). These findings establish a foundation for leveraging trained immunity as a therapeutic strategy for IBD, highlighting the potential of BG in modulating intestinal immunity and restoring gut homeostasis.

### Limitations

Although our reanalysis of publicly available ATAC-seq datasets (data available from CRA014389) targeting monocytes confirmed BG-induced chromatin accessibility enrichment at antibacterial defense-related genomic regions, direct mechanistic evidence for epigenetic and metabolic profiling in our colitis model remains lacking, and we plan to pursue as future work. Moreover, the functional role of CX3CR1+ macrophages remained correlative. Despite transcriptional evidence of repair-related gene expression and phenotypic correlation with colitis resolution, we lack direct causal data due to the unavailability of specific tools for this subset. The fate of adoptively transferred monocytes was not directly traced. Current evidence remains correlative and requires validation through future lineage-tracing studies.

## Materials and methods

### Animals

C57BL/6 J wild-type male mice (6–8 weeks) were purchased from Xiamen University Laboratory Animal Center. *Rag1-/-* and CX3CR1-GFP mice were generously provided by Dr. Kairui Mao and Dr. Xiaofen Chen from Xiamen University, respectively. *Ccr2-/-* mice were kindly provided by Dr. Shih-Chin Cheng from Xiamen University. Mice were maintained under specific pathogen-free conditions. All animal protocols were reviewed and approved by the Institutional Animal Care and Use Committee of Xiamen University.

### In vivo beta-glucan (BG) training and DSS colitis model

Mice were i.p. injected with one dose of BG (1 mg) on day −7/–49. Then, Mice received 3% indicated DSS (36,000–50,000 molecular weight; MP Biomedicals) in their drinking water for 5 days, followed by 2 days or 7 days of distilled water without DSS. Control animals received distilled water for the entire period. Mice were monitored every day from day 0 for body weight until day 7 or 12.

### Histology

Colons were collected from sacrificed mice at the end of each treatment schedule, as described above, colon was fixed with 4% paraformaldehyde and embedded in Paraffin. Tissue sections (5 μm) were prepared, deparaffinized, and stained with hematoxylin and eosin. With minor modifications, by assessing mucosa thickening, inflammatory cells, and submucosa cell infiltration. Each criterion was scored as 0–4, and the sum of each score was defined as the histological score. Histological scores were assigned by experimenters 'blinded' to sample identity.

### Coloscopy

A coloscopy system (ENDOCAM Logic HD, RICHARD WOLF) was used to monitor the severity of DSS-induced colitis. After an 8 hr fast, we administered isoflurane (RWD, Co. Ltd.) and applied glycerin to the anus for smooth insertion.

### In vivo intestine permeability assay

Age and sex-matched mice were orally administered with 0.6 mg/g body weight of a 100 mg/ml solution of FITC-dextran (FD4, Sigma). 5 hr later, retro-orbital blood was collected from each mouse. Serum was prepared by allowing the blood to clot by leaving it undisturbed overnight at 4 °C and then subsequently centrifuged at 5000 rpm for 10 min. Dilutions of FITC-dextran in PBS and separately in pooled mouse serum were used as a standard curve. Absorbance of 50 μl serum was measured at

microplate reader with excitation and emission filters set at 490 and 530 nm, respectively. Experiments were performed at least two independent times each in triplicate.

## RNA extraction and real-time PCR

Total RNA was extracted from colon tissue previously washed from luminal content. Samples were homogenized in TRIZOL. Homogenized tissues were then added with chloroform in order to separate RNA from genomic DNA and proteins. RNA isolation and purification required isopropanol (RNA precipitation) and 70% ethanol (RNA wash). Total RNA was resuspended in Nuclease-free water. Subsequently, 1 µg of RNA was reverse-transcribed using the Hifair ll first Strand cDNA Synthesis SuperMix for qPCR (gDNA digester plus) following manufacturer's instructions. cDNA was analysed through quantitative Real-Time PCR.

## Isolation of cells from lamina propria

To isolate lamina propria mononuclear cells (lamina propria MCs), extraintestinal fat tissue was carefully removed and colons were then flushed of their luminal content with physiologic solution, opened longitudinally and cut into 1 cm pieces. Epithelial cells and mucus were removed by incubation with 10 mL of D-Hank's balanced salt solution (containing EDTA and DTT, free of $Ca^{2+}$ and $Mg^{2+}$) for 20 min at 37 °C at 200 rpm in a constant temperature shaker. Colon pieces were then digested in Hank's (with $Ca^{2+}$ and $Mg^{2+}$) containing 0.5 mg/ml Collagenase IV and 15 µg/ml DNase I, for 40 min at 37 °C, shaking at 200 rpm. The remaining cells were centrifuged and resuspended in FACS buffer (1% FBS, 2 mM EDTA in PBS).

## Bone marrow transplantation

Wild-type CD45.2 mice were subjected to 8 Gy irradiation (RAD SOURCE, RS2000). These irradiated recipients were then divided into three groups: a sham-transplant control group that received an intravenous injection of PBS alone, a control BMT group that received bone marrow cells from naïve CD45.1 donors, and a BG-trained BMT group that received bone marrow cells from CD45.1 donors pretreated with β-glucan. Bone marrow cells were harvested from the femurs of CD45.1 wild-type or BG-trained donor mice. A total of $1\times10^7$ bone marrow cells from the respective donors were transferred to the irradiated CD45.2 recipients via tail-vein injection. Successful engraftment was confirmed by flow cytometric analysis of donor-derived (CD45.1$^+$) cells in peripheral blood (Fig. S5C). Six weeks post-transplantation, mice were administered 2.5% DSS in drinking water to induce colitis and were monitored daily until day 12.

## Adoptive transfer of bone marrow monocytes

BG trained mice were sacrificed, and the femurs were used to harvest bone marrow cells. Bone marrow cells were stained with the following panel: Biotin antibody (B220, CD4, CD8, NK1.1, Ly6G), Live/dead (Cat. L34976, Invitrogen), CD45 (Cat. 45-0451-82, Clone. 30-F11, ebioscience), CD11b (Cat. 48-0112-82, Clone. M1/70, ebioscience), Ly6C (Cat. 128016, Clone. HK1.4, Biolegend). Biotin-labeled lineage antibodies were used to stain bone marrow cells at 4°C for 20 min and then incubated with fluorescence-conjugated streptomycin together with other surface protein antibodies for another 20 min. Ly6C$^{hi}$ monocytes were sorted to >97% purity using a BD Aria III cell sorter.

## RNA-seq

RNA was extracted from colon LPLs of individual mice. RNA integrity was assessed using the RNA Nano 6000 Assay Kit of the Bioanalyzer 2100 system (Agilent Technologies, CA, USA). The clustering of the index-coded samples was performed on a cBot Cluster Generation System using TruSeq PE Cluster Kit v3-cBot-HS (Illumina) according to the manufacturer's instructions. The library was sequenced on an Illumina Novaseq platform and 150 bp paired-end reads were generated.

## scRNA-seq

Isolated lamina propria mononuclear cells, sample from three mice were pooled together and analyzed per condition. For scRNA-seq of Colon CD45$^+$ cells were sorted by BD Aria III. These cells were barcoded with 10x Cellplex oligos before being encapsulated using the 10 X Chromium 3' Reagent Kits v3 according to the manufacturer's instructions.

## RNA-seq data processing and analysis

Raw data of fastq format were quality checked through fastqc software. Gencode and gene model annotation files were downloaded from genome (mm10) website directly. Index of the gencode was built using Hisat2 v2.1.0 and paired-end clean reads were aligned to the gencode using Hisat2 v2.1.0. For differential expression analysis, raw count data (generated via featureCounts) were used as input for DESeq2, with differential genes filtered by |fold change|≥2 and adjusted $p<0.05$. FPKM (Fragment per kilobase per million) values were only used for visualization purposes (e.g. heatmaps, clustering analyses), not for between-sample statistical comparisons. Weighted correlation network analysis (WGCNA) was perform using an R package from Horvath. All differentially expressed genes were plotted with the 'pheatmap' and 'ggplot2' R packages. For KEGG enrichment analysis, $p$-value <0.05 was used as a threshold to determine significant enrichment for gene sets with the 'clusterProfiler' R package.

## scRNA-seq data processing and analysis

Raw sequencing data was processed and aligned 10 mm mouse reference genome with CellRanger (10×Genomics) v 7.2.0. Resulting filtered matrices (count matrices) of molecular counts were used as input for further processing with Seurat package V5.0.1 running under R Studio. First, quality control was performed to create Seurat object with min features >200 and removal of cells having <200 or >8000 expressed genes or >5% mitochondrial counts. The total number of recovered cells was mentioned before. Variable features using FindVariableFeatures (using RNA and vst as an assay and selection method as parameters) and normalization using normalization.method = 'LogNormalize,' scale.factor=10,000 were performed. This integrated data is scaled using ScaleData function using all genes and 'RNA' as assay method.

Doublet cells were filtered by DoubletFinder v3. Principal Component Analysis was performed on variable features using RunPCA and first 30 PCs were chosen. The nearest neighbors using the FindNeighbors(dims = 1:12) and FindClusters(resolution = 0.8) function. For visualization, the non-linear dimensional reduction, such as UMAP analysis was using RunUMAP function from Seurat. To calculate pseudotime based on the Mono/Marco subset data using the monocle3.

## ATAC-seq data analysis

The ATAC-seq data were obtained from the CNCB (China National Center for Bioinformation) under accession number CRA014389. Quality control was performed using FastQC. The paired-end reads were then aligned to the mouse reference genome (mm10) using Bowtie2 with the parameters: -k 1 -D 20 R 3 N 1 L 20 -i S,1,0.50 -X 2000. The aligned reads were converted to sorted BAM files using Samtools. Peak calling was conducted with MACS2 using a q-value threshold of 0.05, a model fold range of 5–50, and the 'keep-dup all' option to retain all tags. Peak quantification was performed using Bedtools and normalized to reads per kilobase per million mapped reads (RPKM). Visualization files in BigWig format were generated using Deeptools. The identified peaks were annotated using the ChIPseeker package, and pathway analysis was carried out with the ClusterProfiler package. Gene Set Enrichment Analysis (GSEA) was performed using the MSigDB database (version v2022.1.Mm).

## Statistical analysis

Statistical analyses were conducted using GraphPad Prism version 9.0 and R statistical software (version 4.2.2, R Foundation for Statistical Computing). Data were analyzed using a two-tailed Student's $t$-test. Multiple-group comparisons were performed using one-way or two-way analysis of variance (ANOVA) followed by Tukey's multiple-comparisons test. Survival data were assessed using Kaplan-Meier survival plots, followed by the log-rank test. Significant differences were indicated by an asterisk ($p<0.05$).

## Acknowledgements

We thank the staff of Xiamen University Laboratory Animal Center. Special thanks to Prof. Kairui Mao and Prof. Xiao-fen Chen for generously providing mice. We appreciate the help of Dr. Jia Zhang and other staff from Prof. Shih-Chin Cheng's laboratory during the study. This project was supported by National Key R&D Program of China (2022YFA1304000), National Natural Science Foundation of

China (82300630, 32161133020), Healthcare System Youth Backbone Talent Training Project of Fujian Province (2023GGB09), Municipal Natural Science Foundation of Xiamen (3502Z20227271), Foundation of State Key Laboratory of Vaccines for Infectious Diseases, Xiang An Biomedicine Laboratory (2023XAKJ0101012), Medical and Health Key Project of Xiamen (3502Z20204007).

## Additional information

### Funding

| Funder | Grant reference number | Author |
| --- | --- | --- |
| National Key R&D Program of China | 2022YFA1304000 | Hongzhi Xu |
| National Natural Science Foundation of China | 82300630 | Shih-Chin Cheng |
| Healthcare System Youth Backbone Talent Training Project of Fujian Province | 2023GGB09 | Hongzhi Xu |
| Municipal Natural Science Foundation of Xiamen | 3502Z20227271 | Hongzhi Xu |
| Foundation of State Key Laboratory of Vaccines for Infectious Diseases, Xiang An Biomedicine Laboratory | 2023XAKJ0101012 | Hongzhi Xu |
| Medical and Health Key Project of Xiamen | 3502Z20204007 | Hongzhi Xu |
| National Natural Science Foundation of China | 32161133020 | Shih-Chin Cheng |

The funders had no role in study design, data collection and interpretation, or the decision to submit the work for publication.

### Author contributions

Yinyin Lv, Data curation, Formal analysis, Validation, Visualization, Methodology, Writing – original draft; Yanyun Fan, Qiongyun Chen, Visualization, Writing – original draft; Qingxiang Gao, Formal analysis, Visualization, Writing – original draft; Yiqun Hu, Lin Wang, Huaxiu Shi, Ermei Chen, Qinyu Xu, Investigation; Ying Cai, Dan Du, Investigation, Methodology; Qingqi Fan, Formal analysis; Linying Li, Methodology; Jianlin Ren, Conceptualization, Resources, Project administration, Writing – review and editing; Shih-Chin Cheng, Funding acquisition, Project administration, Writing – review and editing; Hongzhi Xu, Conceptualization, Funding acquisition, Project administration, Writing – review and editing

### Author ORCIDs

Yinyin Lv https://orcid.org/0000-0002-6371-7418
Shih-Chin Cheng https://orcid.org/0000-0003-1251-8774
Hongzhi Xu https://orcid.org/0000-0002-2725-5359

### Ethics

All animal protocols were reviewed and approved by the Institutional Animal Care and Use Committee of Xiamen University.

Reviewer #1 (Public review): https://doi.org/10.7554/eLife.107339.3.sa1
Reviewer #2 (Public review): https://doi.org/10.7554/eLife.107339.3.sa2
Reviewer #3 (Public review): https://doi.org/10.7554/eLife.107339.3.sa3
Author response https://doi.org/10.7554/eLife.107339.3.sa4

## Additional files

### Supplementary files
MDAR checklist

Supplementary file 1. Flow cytometery antibody panel list.

Supplementary file 2. Primer sequences.

### Data availability
All data relevant to the study are included in the manuscript or uploaded as online supplemental information. Sequencing data have been deposited to the GEO with accession number GSE285859 (single cell RNA-seq), GSE285860 (RNA-seq).

The following datasets were generated:

| Author(s) | Year | Dataset title | Dataset URL | Database and Identifier |
|---|---|---|---|---|
| Lv Y, Chen Q, Gao Q | 2025 | Single cell RNA-seq of CD45+ cell from DSS induced colitic | https://www.ncbi.nlm.nih.gov/geo/query/acc.cgi?acc=GSE285859 | NCBI Gene Expression Omnibus, GSE285859 |
| Lv Y, Chen Q, Gao Q | 2026 | RNA-seq of colon | https://www.ncbi.nlm.nih.gov/geo/query/acc.cgi?acc=GSE285860 | NCBI Gene Expression Omnibus, GSE285860 |

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
