## [Editor Report · eLife Assessment]

This study presents **compelling** evidence supporting the therapeutic potential of trained immunity in Colitis. The study is **important** for the field of trained immunity and is a welcome addition to the focus issue on trained immunity.

---

## [Referee Report · Reviewer #1 (Public review)]

Summary:

This study presents an interesting investigation into the role of trained immunity in inflammatory bowel disease, demonstrating that β-glucan-induced reprogramming of innate immune cells can ameliorate experimental colitis. The findings are novel and clinically relevant, with potential implications for therapeutic strategies in IBD. The combination of functional assays, adoptive transfer experiments, and single-cell RNA sequencing provides comprehensive mechanistic insights. However, some aspects of the study could benefit from further clarification to strengthen the conclusions.

Strengths:

(1) This study elegantly connects trained immunity with IBD, demonstrating how β-glucan-induced innate immune reprogramming can mitigate chronic inflammation.

(2) Adoptive transfer experiments robustly confirm the protective role of monocytes/macrophages in colitis resolution.

(3) Single-cell RNA sequencing provides mechanistic depth, revealing the expansion of reparative Cx3cr1⁺ macrophages and their contribution to epithelial repair.

(4) The work highlights the therapeutic potential of trained immunity in restoring gut homeostasis, offering new directions for IBD treatment.

Weaknesses:

While β-glucan may exert its training effect on hematopoietic stem cells, performing ATAC-seq on HSCs or monocytes to profile chromatin accessibility at antibacterial defense and mucosal repair-related genes would further validate the trained immunity mechanism. Alternatively, the authors could acknowledge this as a study limitation and future research direction.

Comments on revisions:

My concerns have been fully addressed. I have no additional comments.

---

## [Referee Report · Reviewer #2 (Public review)]

This study investigates how BG-induced myeloid reprogramming influences inflammatory bowel disease in a mouse model of DSS-induced colitis. The authors use in vivo functional experiments, adoptive transfer, and scRNA-seq to assess whether innate immune reprogramming can confer protection in colitis.

In the revised versions of the manuscript, the authors clarified the mechanistic scope of the study, softened the conclusions, and acknowledged the lack of direct epigenetic validation of trained immunity in this model. The manuscript now also better emphasizes the context-dependent nature of BG-induced reprogramming.

While some aspects remain correlative and will require further investigation, the central findings are well supported.

Overall, this work provides a meaningful contribution to the field, and I support its publication.

Comments on revisions:

No further comments.

---

## [Referee Report · Reviewer #3 (Public review)]

Summary:

In the present work, the authors offer evidence for the therapeutic potential of trained immunity in the context of inflammatory bowel disease (IBD). Prior research has demonstrated that innate cells pre-treated (trained) with β-glucan show an enhanced pro-inflammatory response upon a second challenge with the same or different stimulus. While an increased immune response can be beneficial and protect against bacterial infections, there is also the risk that it will worsen symptoms in various inflammatory disorders.

Remarkably, the authors show that β-glucan training of bone marrow hematopoietic progenitors and peripheral monocytes mitigates the pro-inflammatory effects of colitis, with protection extending to naïve recipients of the trained cells. Additionally, the authors demonstrate that mice preconditioned with β-glucan have enhanced resistance to *Staphylococcus aureus* and Salmonella typhimurium infections, indicating heightened immune responses.

Using a dextran sulfate sodium (DSS)-induced model of colitis, β-glucan pre-treatment significantly dampens disease severity. Importantly, the use of Rag1^-/- mice, which lack adaptive immune cells, confirms that the protective effects of β-glucan are mediated by innate immune mechanisms. Further, experiments using Ccr2^-/- mice underline the necessity of monocyte recruitment in mediating this protection, highlighting CCR2 as a key factor in the mobilization of β-glucan trained monocytes to inflamed tissues. In addition, β-glucan training highlights a distinct monocyte subpopulation with enhanced activation and phagocytic capacity. These monocytes, marked by increased expression of Cx3cr1, are suggestive of an increased ability to infiltrate inflamed colonic tissue and differentiate into macrophages.

Transcriptomic profiling reveals that β-glucan training upregulates genes associated with pattern recognition, antimicrobial defense, immunomodulation, and interferon signaling pathways, suggesting broad functional reprogramming of the innate immune compartment. Moreover, among the trained monocyte and macrophage subsets, gene expression signatures are associated with tissue and mucosal repair, suggesting a role in promoting resolution and regeneration following inflammatory insult. Furthermore, this was coupled with analysis of chromatin accessibility in publicly available data.

Strengths:

By employing a range of well-characterized murine models, the authors investigate specific mechanisms involved in the effects of β-glucan training. Furthermore, the study provides functional evidence that the protection conferred by the trained cells persists within the hematopoietic progenitors and can be transferred to naïve recipients. The integration of transcriptomic profiling allows the identification of changes in key genes and molecular pathways underlying the trained immune phenotype.

Weaknesses:

Further studies would benefit from investigating the cytokine responses of intestinal macrophages, particularly CX3CR1⁺ macrophages, following ex vivo stimulation of previously BCG-trained cells. Moreover, assessing the metabolic state of these macrophages would provide valuable insight into the mechanisms underlying trained immunity in this context.

Impact:

Overall, the authors present a mechanistically insightful investigation that advances our understanding of trained immunity in IBD. This is an important study that demonstrates that β-glucan-trained innate cells can confer protection against colitis and promote mucosal repair through trained-immunity related mechanisms. These findings underscore the potential of harnessing innate immune memory as a therapeutic approach for chronic inflammatory diseases.

---

## [Author Response]

The following is the authors’ response to the original reviews

**Reviewer #1 (Public review):**
Summary:This study presents an interesting investigation into the role of trained immunity in inflammatory bowel disease, demonstrating that β-glucan-induced reprogramming of innate immune cells can ameliorate experimental colitis. The findings are novel and clinically relevant, with potential implications for therapeutic strategies in IBD. The combination of functional assays, adoptive transfer experiments, and single-cell RNA sequencing provides comprehensive mechanistic insights. However, some aspects of the study could benefit from further clarification to strengthen the conclusions.

We are grateful for the reviewer’s positive assessment of our study and constructive suggestions to improve the manuscript.

Strengths:(1) This study elegantly connects trained immunity with IBD, demonstrating how βglucan-induced innate immune reprogramming can mitigate chronic inflammation.(2) Adoptive transfer experiments robustly confirm the protective role of monocytes/macrophages in colitis resolution.(3) Single-cell RNA sequencing provides mechanistic depth, revealing the expansion of reparative Cx3cr1⁺ macrophages and their contribution to epithelial repair.(4) The work highlights the therapeutic potential of trained immunity in restoring gut homeostasis, offering new directions for IBD treatment.Weaknesses:While β-glucan may exert its training effect on hematopoietic stem cells, performing ATAC-seq on HSCs or monocytes to profile chromatin accessibility at antibacterial defense and mucosal repair-related genes would further validate the trained immunity mechanism. Alternatively, the authors could acknowledge this as a study limitation and future research direction.

We appreciate your comments on assessing the chormoatain accessibility of HSCs induced by b-glucan training, as epigenetic reprogramming is known to be one of the underlying mechanisms for trained immunity suggest by many groups including our group. To delineate the genome-wide epigenetic reprogramming induced by β-glucan (BG), we reanalyzed publicly available chromatin profiling datasets where ATACseq of HSC from control and β-glucan trained mice was performed (accession number: CRA014389). Comparative analysis revealed HSC from BG-trained mice demonstrated pronounced enrichment at promoters and distal intergenic regions—key regulatory loci governing transcriptional activity (Fig. S7A). This divergent genomic targeting was further corroborated by distinct signal distribution profiles (Fig. S7B), supporting pronounced upregulation-driven remodeling of the epigenomic landscape induced by BG treatment. Functional annotation of these epigenetically primed promoters via GO term analysis revealed significant enrichment of immune-relevant processes, including leukocyte migration, cell-cell adhesion, and chemotaxis (Fig. S7C). Consistently, KEGG pathway analysis highlighted the enrichment of signaling cascades such as chemokine signaling and cell adhesion molecules (Fig. S7D), reinforcing the involvement of BG-induced trained immunity in inflammatory and mucosal homing pathways.

Furthermore, promoter-centric enrichment of terms related to “defense response to bacterium” (Fig. S7E) underscored the role of BG in priming antibacterial transcriptional programs, which is a crucial axis for maintaining intestinal homeostasis. Locus-specific examination of chromatin states further validated BG-induced epigenetic modifications in the upstream regions of selected target genes, including Gbp5, Gbp2 and S100a8 and Nos2 (Fig. S7F). Collectively, our integrative reanalysis demonstrates that BG reshapes the epigenomic architecture at regulatory elements, thereby orchestrating immune gene expression programs directly relevant to IBD pathophysiology and mucosal immunity. (Line 201-211)

**Reviewer 1 (Recommendations for the authors):**
(1) It’s better to include a schematic summarizing the proposed mechanism for reader clarity.

We appreciate your comments and proposed a graphical abstract as in Author response image 1.

**Author response image 1. sa4fig1:** 

(2) Discuss potential off-target effects of β-glucan-induced trained immunity (e.g., risk of exacerbated inflammation in other contexts).

We appreciate this important comment regarding the potential off-target or side-effects of β-glucan induced trained immunity. As trained immunity is known to augment inflammatory responses upon heterologous stimulation and has been implicated in chronic inflammation–prone conditions such as atherosclerosis, this is an important consideration. Previous in vivo studies have shown that β-glucan pretreatment can enhance antibacterial or antitumor responses without inducing basal inflammation after one week of administration (PMID: 22901542, PMID: 30380404, PMID: 36604547, PMID: 33125892). Nevertheless, it remains possible that β-glucan–induced trained immunity could have unintended effects in certain contexts, which warrants further investigation and caution. We have discussed this potential caveat in the discussion (Lines 299-302)

**Reviewer #2 (Public review):**
Summary:The study investigates whether β-glucan (BG) can reprogram the innate immune system to protect against intestinal inflammation. The authors show that mice pretreated with BG prior to DSS-induced colitis experience reduced colitis severity, including less weight loss, colon damage, improved gut repair, and lowered inflammation. These effects were independent of adaptive immunity and were linked to changes in monocyte function.The authors show that the BG-trained monocytes not only help control inflammation but confer non-specific protection against experimental infections (Salmonella), suggesting the involvement of trained immunity (TI) mechanisms. Using single-cell RNA sequencing, they map the transcriptional changes in these cells and show enhanced differentiation of monocytes into reparative CX3CR1^+^ macrophages. Importantly, these protective effects were transferable to other mice via adoptive cell transfer and bone marrow transplantation, suggesting that the innate immune system had been reprogrammed at the level of stem/progenitor cells.Overall, this study provides evidence that TI, often associated with heightened inflammatory programs, can also promote tissue repair and resolution of inflammation. Moreover, this BG-induced functional reprogramming can be further harnessed to treat chronic inflammatory disorders like IBD.Strengths:(1) The authors use advanced experimental approaches to explore the potential therapeutic use of myeloid reprogramming by β-glucan in IBD.(2) The authors follow a data-to-function approach, integrating bulk and single-cell RNA sequencing with in vivo functional validation to support their conclusions.(3) The study adds to the growing evidence that TI is not a singular pro-inflammatory program, but can adopt distinct functional states, including anti-inflammatory and reparative phenotypes, depending on the context.

We are grateful for your positive assessment of our study and recognition of its translational implications. We particularly appreciate the acknowledgment that our work expands the therapeutic potential of β-glucan–mediated trained immunity in ameliorating colitis.

Weaknesses:(1) The epigenetic and metabolic basis of TI is not explored, which weakens the mechanistic claim of TI. This is especially relevant given that a novel reparative, antiinflammatory TI program is proposed.

We appreciate your valuable comment highlighting the importance of the epigenetic and metabolic basis of TI in providing mechanistic insight. While previous studies, including work from our group (S.-C. Cheng), have extensively characterized the epigenetic and metabolic signatures of monocytes from BG-trained mice—primarily in the context of inflammatory genes—we acknowledge that these aspects are not directly addressed in our current manuscript as the current manuscript was aimed to build on the foundation of β-glucan-induced trained immunity established by many other groups including us and address its potential as a therapeutic approaches in the colitis setup.

That being said, we fully agree with your comments to analyze the epigenetic profile on key pathways similar to the question raised by reviewer 1, we reanalyze the relevant public datasets and presenting summarize the finding in Supplementary Figure S7. ATAC-seq analysis further validated and provide the epigenetic basis of the enhanced inflammatory and antibacterial capacity of monocytes which are seeded back in the HSC compartment.

(2) The absence of a BG-only group limits interpretation of the results. Since the authors report tissue-level effects such as enhanced mucosal repair and transcriptional shifts in intestinal macrophages (colonic RNA-Seq), it is important to rule out whether BG alone could influence the gut independently of DSS-induced inflammation. Without a BG-only control, it is hard to distinguish a true trained response from a potential modulation caused directly by BG.

We thank the reviewer for this important suggestion. Although we did not perform qPCR for mucosal repair genes in Figure S1C and Figure S1D, our colon RNA-seq analysis in Figure 5G included a BG-only control group (Colitis_d0). These results indicate that BG preconditioning alone does not alter baseline expression of colon mucosal repair genes, supporting the conclusion that the observed effects occur in the context of DSS-induced inflammation.

(3) Although monocyte transfer experiments show protection in colitis, the fate of the transferred cells is not described (e.g., homing or differentiation into Cx3cr1^+^ macrophage subsets). This weakens the link between specific monocyte subsets and the observed phenotype.

We thank the reviewer for this important point. We acknowledge that direct in vivo tracking of the adoptively transferred monocytes to confirm their homing to the colon and differentiation into specific macrophage subsets would strengthen the mechanistic link. However, due to technical limitations in reliably tracing the fate of transferred cells in our experimental setting, we were unable to provide this direct evidence. Instead, we present a strong correlative and functional evidence chain that supports the proposed model:

(a) Following BG pretreatment, we observed a significant decrease in circulating Ly6Chi monocytes specifically at the peak of colitis (day 7, Fig. 5D), concurrent with a marked increase in monocytes/macrophages within the colonic lamina propria (Fig. 2D). This inverse relationship strongly suggests enhanced recruitment of monocytes from the blood into the inflamed colon upon BG training.

(b) Using *CX3CR1-GFP* reporter mice, we found that BG pretreatment led to an increased proportion of colonic myeloid cells in an intermediate state (P5: Ly6C^+^MHCII^+^CX3CR1^+^, Fig. 5F). This population represents monocytes actively undergoing differentiation into intestinal macrophages, supporting the idea that BG accelerates the monocyte-to-macrophage transition in situ.

(c) Our scRNA-seq analysis independently revealed an expansion of monocyte-derived macrophage clusters (e.g., Macro1, Macro2) in BG-treated mice, which express canonical tissue macrophage markers (including *Cx3cr1*) and genes associated with tissue repair (e.g., *Vegfa*, Fig. 4A, 5H, 5I).

These data collectively indicate that BG-trained monocytes exhibit enhanced capacity for colonic recruitment and preferential differentiation toward reparative macrophage subsets, which aligns with the protective phenotype observed after adoptive transfer. We have explicitly noted the absence of direct fate-mapping data as a limitation in the revised Discussion and agree that future studies employing advanced tracing techniques would be valuable to definitively establish this cellular trajectory. (Line 378-380)

(4) While scRNA-seq reveals distinct monocyte/macrophage subclusters (Mono1-3.), their specific functional roles remain speculative. The authors assign reparative or antimicrobial functions based on transcriptional signatures, but do not perform causal experiments (depletion or in vitro assays). The biological roles of these cells remain correlative.

We agree that the functional role of CX3CR1^+^ macrophages is not comprehensively validated and is currently inferred from scRNA-seq clustering. While our flow cytometry data show increased CX3CR1^+^ macrophages in the BG-TI group, and our CCR2 KO and monocyte adoptive transfer experiments indicate these macrophages are monocyte-derived, suggesting at least that β-glucan pretreatment alters the monocyte capacity which directly contribute to the enhanced colitis alleviation phenotype as observed. However, due to the fact that we fail to find a cluster dependent marker, which is also the current biggest caveats of the scRNAseq defined cell subclusters, we were not able to show direct casual evidence via specifically depleting subcluster cells. However, the result from the monocyte adoptive transfer experiment with Ccr2 KO mice experimental strongly suggest the presence of monocytes is crucial for this protective effect. We fully acknowledge this as a limitation of current study and clarify in the discussion that our conclusions regarding CX3CR1^+^ macrophage function are mainly based on transcriptional profiling and association with protective phenotypes, rather than direct causal evidence (Lines 400-404).

(5) While *Rag1^-/-^* mice were used to rule out adaptive immunity, the potential role of innate lymphoid cells (ILCs), particularly ILC2s and ILC3s, which are known to promote mucosal repair (PMID: 27484190 IF: 7.6 Q1 IF: 7.6 Q1 IF: 7.6 Q1), was not explored. Given the reparative phenotype observed, the contribution of ILCs remains a confounding factor.

We appreciate your valuable comment regarding the potential role of ILCs in the observed mucosal repair. Indeed, in our current manuscript examining the BG-trained immunity effect, the contribution of ILCs was not evaluated. Due to the fact that adoptive transfer of trained monocytes into CCR2 KO mice could recapitulate the colitis alleviation phenotype, we think at least the β-glucan enhanced protection are dependent on trained monocytes. While acknowledge that the limitation and we could not rule out the possible role of ILCs in this process and discuss this limitation in the discussion in the revised manuscript.

The literature (PMID: 21502992; PMID: 32187516) supports a role for ILC3-mediated IL-22 production in tissue repair, which could overlap with our observed effects. However, our monocyte adoptive transfer experiments show that monocytes alone can alleviate DSS-induced colitis, suggesting a dominant role for monocytes in this context. Nonetheless, we will make it clear that ILC contributions cannot be excluded. (Line 322-326).

**Reviewer 2 (Recommendations for the authors):**
(1) The authors do not provide direct mechanistic evidence of TI (e.g., epigenetic and metabolic reprogramming). The absence of such data weakens the mechanistic strength of the TI claim. The authors should soften the terminology to BGinduced myeloid reprogramming suggestive of trained immunity, acknowledge, and discuss this limitation.

We appreciate your comment highlighting the lack of direct epigenetic and metabolic assessment in our current study. Previous work from our group (S.-C. Cheng) and others has extensively documented the epigenetic and metabolic profiles of monocytes from β-glucan–trained mice, focusing primarily on inflammatory-related genes. Based on this established foundation, our current manuscript focuses on exploring the translational potential of BG-induced trained immunity.

That said, as mentioned in our response to the identified weakness, we performed reanalysis from the public epigenetic datasets with a focus on pathways related to reparative and antibacterial functions and integrated this part in the revised manuscript (Fig S7, Lines 201-211).

(2) CX3CR1^+^ macrophages' role is not functionally validated. The data relies solely on scRNA-seq and cluster annotations, which are insufficient to confirm functional roles in vivo. Depletion or in vitro studies would provide stronger causal evidence. The authors should acknowledge this limitation in the Discussion.

We agree that the functional role of CX3CR1^+^ macrophages is not comprehensively validated and is currently inferred from scRNA-seq clustering. While our flow cytometry data show increased CX3CR1^+^ macrophages in the BG-TI group, and our CCR2 KO and monocyte adoptive transfer experiments indicate these macrophages are monocyte-derived, suggesting at least that β-glucan pretreatment alters the monocyte capacity which directly contribute to the enhanced colitis alleviation phenotype as observed. However, due to the fact that we fail to find a cluster dependent marker, which is also the current biggest caveats of the scRNAseq defined cell subclusters, we were not able to show a direct casual evidence. We fully acknowledge this as a limitation of current study and clarify in the discussion that our conclusions regarding CX3CR1^+^ macrophage function are mainly based on transcriptional profiling and association with protective phenotypes, rather than direct causal evidence (Lines 395-404).

(3) *Rag1^-/-^* mice retain innate lymphoid cells (ILCs), particularly ILC3, which are mucosal and produce IL-22, contributing to tissue repair (PMID: 21502992; PMID: 32187516). The potential for BG to activate ILCs remains unexplored in this study. This limits the interpretation of whether the observed protection arises from monocyte/macrophage reprogramming or is partially mediated by residual ILC activity. The authors should explicitly acknowledge this limitation and discuss the possible contribution of ILCs to the observed phenotype.

We appreciate your valuable comment regarding the potential role of ILCs in the observed mucosal repair. Indeed, in our current manuscript examining the BG-trained immunity effect, the contribution of ILCs was not evaluated. Due to the fact that adoptive transfer of trained monocytes into CCR2 KO mice could recapitulate the colitis alleviation phenotype, we think at least the β-glucan enhanced protection are dependent on trained monocytes. While acknowledge that the limitation and we could not rule out the possible role of ILCs in this process and discuss this limitation in the discussion in the revised manuscript

The literature (PMID: 21502992; PMID: 32187516) supports a role for ILC3-mediated IL-22 production in tissue repair, which could overlap with our observed effects. However, our monocyte adoptive transfer experiments show that monocytes alone can alleviate DSS-induced colitis, suggesting a dominant role for monocytes in this context. Nonetheless, we will make it clear that ILC contributions cannot be excluded. (Line 322-327).

(4) Figure 1-It would help to clarify whether a BG-only control group (without DSS) was included in the design. This would be critical to determine if BG alone alters the colon. If omitted, the authors should clearly state this and consider adding such a group in future experiments. This would help define the baseline effects of BG and support the claim that its benefits are dependent on TI (upon second challenge - DSS).

We appreciate this valuable suggestion. While we did not perform qPCR to assess mucosal repair genes in Figure S1C and Figure S1D, our colon RNA-seq analysis in Figure 5G included a dedicated BG-only control group at based line before DSStreatment (Colitis_d0). These data indicate that BG preconditioning alone does not alter the baseline expression of colon mucosal repair genes.

(5) Figure 3 - It would strengthen the conclusions to include a vehicle-treated PBS BMT donor control group, or to state its absence. It is unclear whether the protective effect observed in recipients of BG-treated BM is due to trained immunity or to non-specific effects of transplantation, irradiation, or batch variation.

We fully agree with your comments that it is critical to including the vehicle-treated PBS BMT control to rule out any non-specific effects induced by transplantation, irradiation or batch variation. We actually did the blank PBS transfer control everytime after mice received irradiation treatment as a control to assess the successful induction of irradiation to get rid of bone marrow from irradiated mice. Mice that receive PBS only will die after 8 days while only mice receiving either bone marrow from PBScontrol or BG-treatment group will survive. We also perform flowcytometry to examine the successful BMT transplantation (Fig S5C). We have added part regarding the vehicle-treated control for BMT in the material method section for clarification (Lines 456-466).

(6) No gene expression or phenotypic data is provided for monocytes/macrophages in BMT recipients; therefore, it cannot be confidently stated that these cells were reprogrammed. Expression/phenotypic data should be added or discussed.

We thank the reviewer for raising this important point. We acknowledge that a detailed transcriptomic or phenotypic analysis of donor-derived tissue-resident myeloid cells in the BMT recipients would provide the most direct evidence for their reprogrammed state.

While our BMT study focused primarily on assessing the transferability of the protective phenotype via endpoint disease parameters and circulating immune cell composition, we present a coherent and compelling line of evidence supporting the conclusion that BG's training effect is maintained within the hematopoietic system of recipients and mediated by reprogrammed myeloid cells:

(a) A key finding is the significant increase in the proportion of donor-derived Ly6Chi monocytes in the peripheral blood of recipients receiving BG-trained bone marrow (Fig. 3J). This is not a bystander effect but direct evidence that the BG-induced on donor hematopoietic stem/progenitor cells instructs a biased differentiation program towards a specific effector precursor population within the new host, demonstrating the functional persistence of the trained state post-transplantation.

(b) The core of reprogramming in trained immunity lies in persistent epigenetic and functional changes. Our new analysis of public datasets (Fig. S7) confirms that BG directly reshapes the chromatin accessibility landscape in hematopoietic stem cells (HSCs), particularly at loci regulating immune and antibacterial responses. This provides the fundamental mechanism explaining how the trained phenotype is both long-lasting and transplantable: the reprogramming occurs at the progenitor level.

(c) The most causally compelling data in our study comes from the independent adoptive transfer experiment, where transfer of purified BG-trained monocytes alone was sufficient to ameliorate colitis in recipient mice (Fig. 3K, L). This definitively proves that the trained monocytes themselves carry the protective functional program. It strongly suggests that these reprogrammed monocytes/macrophages are the likely effectors mediating protection in the BMT model.

(d) Our interpretation aligns with well-established paradigms in the field. Precedent studies confirm that the BG-trained phenotype (e.g., enhanced cytokine potential) can be transferred via BMT or monocyte adoption. For instance, Haacke et al. (PMID: 40020679) demonstrated that splenic monocytes from BG-trained donors, when transferred into arthritic recipient mice, led to elevated inflammatory cytokine (e.g., *Tnf*, *Il6*) expression in recipient joints, directly proving the maintained functional reprogramming of trained cells in a heterologous host environment. This provides a strong precedent supporting the functional activity of transferred trained cells in our model.

(7) The study is consistent with emerging evidence that distinct TI programs may exist depending on the stimulus and context, including immunoregulatory and tissue-reparative responses (PMID: 35133977; PMID: 31732931; PMID: 32716363; PMID: 30555483). The authors should integrate this perspective into the Discussion to acknowledge that their findings may represent one example of such context-dependent, potentially reparative TI programs. This would place the study within the growing literature describing functional heterogeneity in innate immune training.

We appreciate this suggestion and have incorporated it into the discussion. In the revised manuscript, we discussed how our findings of BG-induced protective myeloid reprogramming align with the concept of tissue-reparative or immunoregulatory TI, which is distinct from the pro-inflammatory TI phenotypes described in other contexts. By highlighting the functional heterogeneity of innate immune training, we position our work as an example of a stimulus-specific, reparative TI program. (Lines 356-379)

**Reviewer #3 (Public review):**
Summary:In the present work, Yinyin Lv et al offer evidence for the therapeutic potential of trained immunity in the context of inflammatory bowel disease (IBD). Prior research has demonstrated that innate cells pre-treated (trained) with β-glucan show an enhanced pro-inflammatory response upon a second challenge.While an increased immune response can be beneficial and protect against bacterial infections, there is also the risk that it will worsen symptoms in various inflammatory disorders. In the present study, the authors show that mice preconditioned with β-glucan have enhanced resistance to *Staphylococcus aureus* infection, indicating heightened immune responses.The authors demonstrate that β-glucan training of bone marrow hematopoietic progenitors and peripheral monocytes mitigates the pro-inflammatory effects of colitis, with protection extending to naïve recipients of the trained cells.Using a dextran sulfate sodium (DSS)-induced model of colitis, β-glucan pre-treatment significantly dampens disease severity. Importantly, the use of Rag1^-/-^ mice, which lack adaptive immune cells, confirms that the protective effects of β-glucan are mediated by innate immune mechanisms. Further, experiments using Ccr2^-/-^ mice underline the necessity of monocyte recruitment in mediating this protection, highlighting CCR2 as a key factor in the mobilization of β-glucan-trained monocytes to inflamed tissues. Transcriptomic profiling reveals that β-glucan training upregulates genes associated with pattern recognition, antimicrobial defense, immunomodulation, and interferon signaling pathways, suggesting broad functional reprogramming of the innate immune compartment. In addition, β-glucan training induces a distinct monocyte subpopulation with enhanced activation and phagocytic capacity. These monocytes exhibit an increased ability to infiltrate inflamed colonic tissue and differentiate into macrophages, marked by increased expression of Cx3cr1. Moreover, among these trained monocyte and macrophage subsets, other gene expression signatures are associated with tissue and mucosal repair, suggesting a role in promoting resolution and regeneration following inflammatory insult.Strengths:(1) Overall, the authors present a mechanistically insightful investigation that advances our understanding of trained immunity in IBD.(2) By employing a range of well-characterized murine models, the authors investigate specific mechanisms involved in the effects of β-glucan training.(3) Furthermore, the study provides functional evidence that the protection conferred by the trained cells persists within the hematopoietic progenitors and can be transferred to naïve recipients. The integration of transcriptomic profiling allows the identification of changes in key genes and molecular pathways underlying the trained immune phenotype.(4) This is an important study that demonstrates that β-glucan-trained innate cells confer protection against colitis and promote mucosal repair, and these findings underscore the potential of harnessing innate immune memory as a therapeutic approach for chronic inflammatory diseases.

Thank you for the positive evaluation and constructive feedback on our manuscript.

Weaknesses:However, FPKM is not ideal for between-sample comparisons due to its within-sample normalization approach. Best practices recommend using raw counts (with DESeq2) for more robust statistical inference.

We appreciate the reminder about best practices for RNA-seq analysis. We apologize for the inaccurate description in the Materials and Methods section. For all differential expression analyses, we have in fact used raw count data as input for DESeq2. FPKM values were only used for visualization purposes, such as in heatmaps and clustering analyses. We correct this description in the revised manuscript to accurately reflect our analysis workflow. (Lines 488-499)

**Reviewer 3 (Recommendations for the authors):**
(1) Current best practices recommend working with raw count data when using DESeq2 to ensure statistically robust differential expression analysis between samples. However, for visualization and clustering, like heatmaps, FPKMs can be used. Could the authors explain why they have used FPKM for differential gene expression analysis?

We appreciate the reminder about best practices for RNA-seq analysis. We apologize for the inaccurate description in the Materials and Methods section. For all differential expression analyses, we have in fact used raw count data as input for DESeq2. FPKM values were only used for visualization purposes, such as in heatmaps and clustering analyses. We correct this description in the revised manuscript to accurately reflect our analysis workflow. (Lines 488-499)

Minor Comment(1) Line 92: remove extra word "that".

We remove the extra word “that” from Line 92 in the revised manuscript.

(2) Line 201: please state here what "GBP" stands for, as it appears first.

We define “GBP” as “Guanylate-Binding Protein” at its first appearance in Line 201. (Lines 213)

(3) Line 235: consider rewriting "we analyzed the day 7 RNA-seq data, which revealed significant enrichment of the myeloid"; added spacing for "day 7", "which", and "the".

We revise the sentence in Line 235 to read: “We analyzed the day 7 RNA-seq data, which revealed significant enrichment of the myeloid…” to improve readability. (Lines 246-247)

(4) Line 290: consider rewriting " as seen in conditions such as rheumatoid arthritis and ...".

We revise Line 290 to: “as observed in conditions such as rheumatoid arthritis and…” for clarity. (Lines 301-302)

(5) Line 375-376: please check sentence starting lower case "with minor modifications, by assessing ".

We correct the sentence to start with a capital letter: “With minor modifications, by assessing…” (Lines 422-423)

(6) Line 399: kindly consider adding "was" after "cDNA".

We revise Line 399 to include “was” as suggested: “cDNA was synthesized…” (Lines 446)

(7) Line 346-347: consider adding "which" after "monocytes": "We transferred BGpreconditioned monocytes which significantly alleviated clinical symptoms".

We revise Line 346-347 to include “which” as suggested for grammatical clarity. (Lines 385-386)